# Accelerated signal propagation speed in human neocortical dendrites

**Gáspár Oláh[1†§], Rajmund Lákovics[1†], Sapir Shapira[2†], Yonatan Leibner[2], Attila Szücs[3], Éva Adrienn Csajbók[1], Pál Barzó[4], Gábor Molnár[1*‡], Idan Segev[2‡], Gábor Tamás[1*‡]**

[1]HUN-REN-SZTE Research Group for Cortical Microcircuits, Department of Physiology, Anatomy and Neuroscience, University of Szeged, Szeged, Hungary; [2]Edmond and Lily Safra center for Brain Sciences, The Hebrew University of Jerusalem, Jerusalem, Israel; [3]Department of Physiology and Neurobiology, Institute of Biology, Eötvös Loránd University, Budapest, Hungary; [4]Department of Neurosurgery, University of Szeged, Szeged, Hungary

**\*For correspondence:**
molnarg@bio.u-szeged.hu (GM);
gtamas@bio.u-szeged.hu (GT)

[†]These authors contributed equally to this work
[‡]These authors also contributed equally to this work

**Present address:** [§]Laboratory of Cellular Neurophysiology, ELKH, Institute of Experimental Medicine, Budapest, Hungary

**Competing interest:** The authors declare that no competing interests exist.

## eLife Assessment

This study provides **valuable** observations indicating that human pyramidal neurons propagate information as fast as rat pyramidal neurons despite their larger size. **Convincing** evidence demonstrates that this property is due to several biophysical properties of human neurons. This study will be of interest to neurophysiologists.

**Abstract** Human-specific cognitive abilities depend on information processing in the cerebral cortex, where the neurons are significantly larger and their processes longer and sparser compared to rodents. We found that, in synaptically connected layer 2/3 pyramidal cells (L2/3 PCs), the delay in signal propagation from soma to soma is similar in humans and rodents. To compensate for the longer processes of neurons, membrane potential changes in human axons and/or dendrites must propagate faster. Axonal and dendritic recordings show that the propagation speed of action potentials (APs) is similar in human and rat axons, but the forward propagation of excitatory postsynaptic potentials (EPSPs) and the backward propagation of APs are 26 and 47% faster in human dendrites, respectively. Experimentally-based detailed biophysical models have shown that the key factor responsible for the accelerated EPSP propagation in human cortical dendrites is the large conductance load imposed at the soma by the large basal dendritic tree. Additionally, larger dendritic diameters and differences in cable and ion channel properties in humans contribute to enhanced signal propagation. Our integrative experimental and modeling study provides new insights into the scaling rules that help maintain information processing speed albeit the large and sparse neurons in the human cortex.

## Introduction

The human neocortex is thought to be one of the most complex biological structures yet most of our knowledge regarding the properties of individual cortical neurons and their synapses is based on experiments performed in model organisms. Recent findings in human specimens indicated the emergence of new cell types in the human neocortex (*Yáñez et al., 2005*; *Berg et al., 2021*; *Boldog et al., 2018*; *Deitcher et al., 2017*; *Oberheim et al., 2009*) and species-related differences in transmitter release probability (*Szegedi et al., 2016*), regenerative dendritic events (*Beaulieu-Laroche et al., 2018*; *Beaulieu-Laroche et al., 2021*; *Gidon et al., 2020*), ion channel composition of the

dendrites (*Kalmbach et al., 2018*), temporal dynamics of synaptic potentiation (*Verhoog et al., 2013*), and activity patterns of the microcircuits (*Komlósi et al., 2012*; *Molnár et al., 2008*; *Szegedi et al., 2016*). Pioneering experiments indicate that human dendrites could evolve in ways favoring mechanisms not yet found in other species (*Beaulieu-Laroche et al., 2018*; *Gidon et al., 2020*) and might contribute to the apparent efficacy of human cognitive performance (*Goriounova et al., 2018*). Functional differences are accompanied by a divergence in morphological features, ranging from general alterations in the thickness of cortical layers to increasing complexity in anatomical properties of classical cell types (*Deitcher et al., 2017*; *Mohan et al., 2015*). Human pyramidal cells with larger and more extensively branching dendritic trees have an opportunity to receive a higher number of synaptic inputs (*Benavides-Piccione et al., 2020*; *Loomba et al., 2022*). This, when combined with the increased morphological complexity, endows human cortical neurons with enhanced computational and encoding capabilities (*Beaulieu-Laroche et al., 2021*; *Deitcher et al., 2017*).

However, the increase in the size of dendrites and axons might come with a cost of longer signal propagation times of both synaptic potentials in dendrites (larger dendritic delay) as well as action potentials in axons (axonal delay). This will slow down information processing, both within individual cortical neurons as well as in respective cortical circuits (*Buzsáki et al., 2013*; *Vetter et al., 2001*). Indeed, transferring large amounts of information within and between brain regions in a short amount of time, and the capability of the neuronal circuit to respond sufficiently fast to its environment, is an important evolutionary function of neuronal networks (*Buzsáki et al., 2013*; *Laughlin and Sejnowski, 2003*). Increased cell-to-cell delay will also affect plasticity/learning processes that depend on the timing between the pre- and the post-synaptic action potentials, e.g. the spike-timing-dependent plasticity (STDP) mechanism. It was, therefore, suggested that certain scaling morphological rules must be applied so that animals with larger brains can still function adequately in their environment (*West et al., 1997*). Is that the case for cortical neurons in humans?

We set out in this study to directly measure the speed of signal propagation in both dendrites and axons of individual human and rat L2/3 pyramidal cells and applied experiments-based models to identify cellular and subcellular properties involved in controlling neuron-to-neuron propagation delays. Our integrative experimental and modeling study provides insights into the scaling rules that enable to preserve of information processing speed albeit the much larger neurons in the human cortex.

## Results

### Signal propagation paths and delays in human and rat pyramid to pyramid connections

We followed recent results indicating differences in the density and size of human and mouse supragranular pyramidal cells (PCs) (*Berg et al., 2021*) in a human-rat setting. As expected, measurements on 3D reconstructions based on randomly selected, electrophysiologically recorded, and biocytin-filled human (n=30) and rat (n=30) L2/3 cortical pyramidal cells (*Figure 1—figure supplement 1A*) show significant differences in the horizontal (463.17±119.48 vs 324.79±80.58 μm, *t*-test: p=1.687 × 10⁻⁶) and vertical extensions (542.58±146.89 vs 409.99±102.69 μm, *t*-test: p=0.00013), and in the total dendritic (9054.94±3699.71 vs 5162.68±1237.71 μm, *t*-test: p=7.203 × 10⁻⁷) and apical dendritic length (4349.76±1638.39 vs 2592.15±818.26 μm, *t*-test: p=1.638 × 10⁻⁶, *Figure 1—figure supplement 1B and C*).

To examine the temporal aspects of information propagation in excitatory microcircuits, we performed simultaneous whole-cell patch clamp recordings in synaptically connected L2/3 PCs from acute neocortical slices from rat and human tissues (*Figure 1*). EPSPs were measured in response to single AP in presynaptic cells (*Figure 1B*). Synaptic latency was calculated as the time difference between the peak of the presynaptic AP and the onset point of the postsynaptic EPSP (see *Figure 1B* and Methods). We did not find significant differences in synaptic latencies between human and rat PC-to-PC connections (rat: 1.126±0.378 ms, rat: n=19, human: 1.111±0.306 ms, n=17, Mann-Whitney test: p=0.949). Both pre- and postsynaptic PCs were filled with biocytin during recordings allowing for post hoc identification of close appositions between presynaptic axons and postsynaptic dendrites (*Frick et al., 2008*; *Figure 1A*). We measured the shortest axonal path lengths linking the presynaptic soma to close appositions on the postsynaptic dendrite (rat: 168.267±49.59 μm, human:

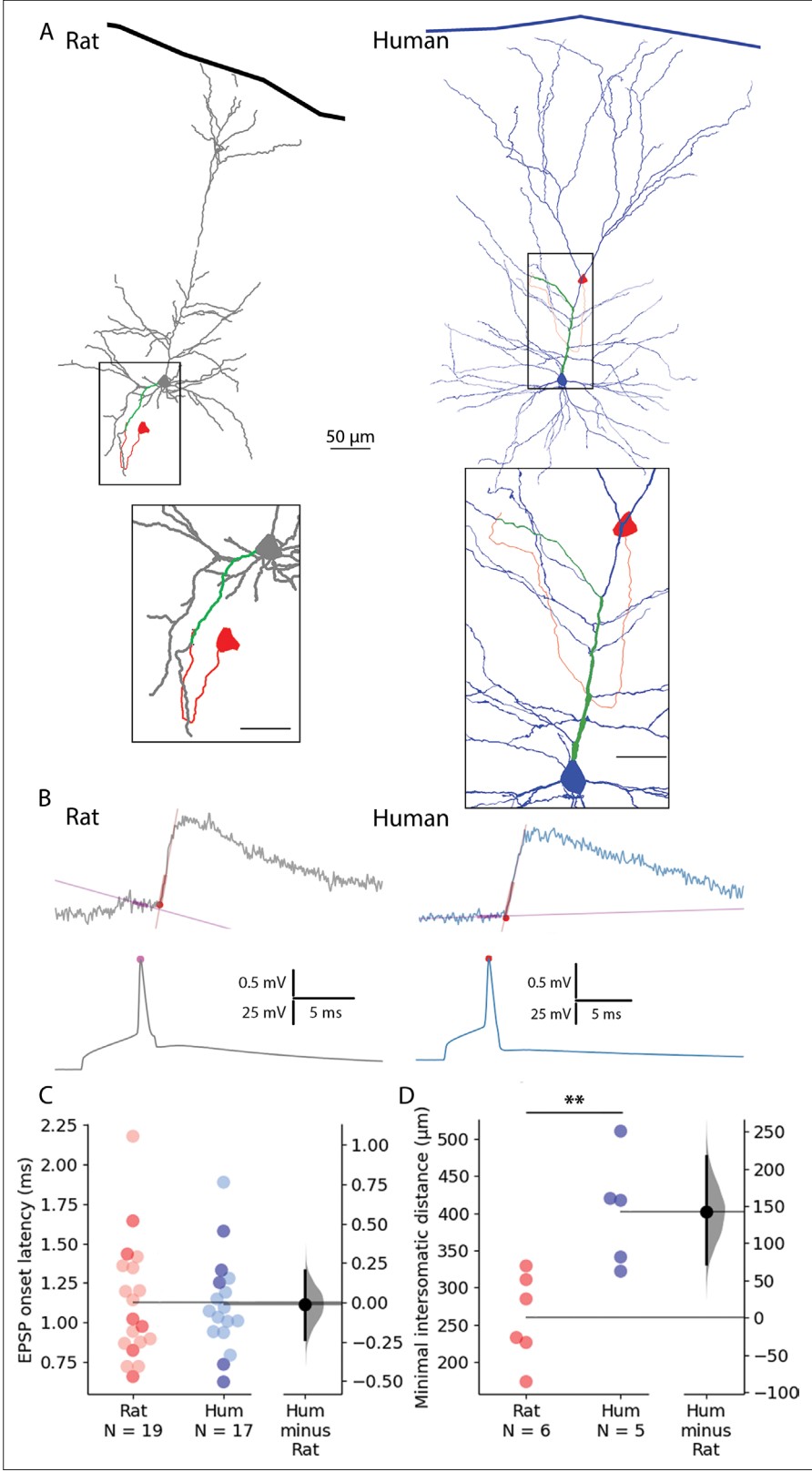

**Figure 1.** Paired recordings from synaptically connected layer 2/3 rat and human pyramidal cells.
(**A**) Representative reconstructions of electrophysiologically recorded and biocytin-filled rat (left, gray soma and dendrites) and human (right, blue soma and dendrites) synaptically connected pyramidal cell pairs. The presynaptic soma and the axon are in red; the postsynaptic dendritic path from the synapse to the soma is

*Figure 1 continued*

highlighted in green. Minimal intersomatic distance was calculated as the sum of the shortest presynaptic axonal (red) and postsynaptic dendritic (green) paths. Boxed region is magnified on the bottom. Scale bars for insets are 20 µm. (**B**) Synaptic latency was determined as the time difference between the peak of the presynaptic AP (pink dot) and the onset of the postsynaptic excitatory postsynaptic potential (red dot). Straight lines indicate baseline and rise phase fitting. (**C**) Summary of synaptic latencies in rat (red) and human (blue) cell pair recordings. Each dot represents the average latency in a cell measured from the action potential (AP) peak to excitatory postsynaptic potentials (EPSP) onset as illustrated in panel B. The darker colors represent the paired recordings with full reconstruction. For these data points there was no significant difference between the two species (Mann-Whitney test: p=0.931). The extended dataset with cell pairs without reconstruction shows no significant difference between the two species (Mann-Whitney test: p=0.949). (**D**) Minimal intersomatic distance of the recorded cell pairs. Intersomatic distance was calculated through every putative synapse and the shortest was taken into account. The minimal intersomatic distance was significantly longer in the human dataset compared to rats (Mann-Whitney test: p=0.009). **p<0.01.

The online version of this article includes the following figure supplement(s) for figure 1:

**Figure supplement 1.** Size comparison of layer 2/3 pyramidal cells in the human and rat cortex.

---

272.22±73.14 µm) and the shortest dendritic path lengths from close appositions found exclusively on dendritic spine heads to the postsynaptic soma (rat: 84.9±18.301 µm, human: 129.48±40.005 µm) in a subset of recordings (rat: n=6, human: n=5). Consequently, we found that the minimal intersomatic distance (the sum of the shortest axonal and dendritic paths) in each synaptically connected PC-to-PC pair was significantly smaller in rats compared to humans (rat: 259.7±58.8 µm, human: 402.12±74.757 µm, Mann-Whitney test: p=0.009, *Figure 1D*). We did not find a significant difference in these paired recordings in synaptic latency (rat: 1.09±0.375ms, n=6 from n=6 rats; human: 1.102±0.408ms, n=5 from n=5 patients; Mann-Whitney test: p=0.931, *Figure 1C*, darker dots). Given that similar synaptic latencies accompany different lengths for signal propagation in the two species, membrane potentials (APs and/or EPSPs) are likely to propagate faster in human PC-to-PC connections.

## Direct measurements of signal propagation in PC dendrites and axons

Compensation of longer axonal and dendritic paths must be explained by the higher velocity of signal propagation along axons and/or dendrites. We, therefore, asked whether interspecies differences can be found in axonal and/or dendritic signal propagation in L2/3 PCs.

First, we investigated whether we could find dissimilarities between the two species in the speed of signal propagation along axons of PCs. We whole-cell recorded the soma and a distal axon simultaneously, positioning the axonal recording electrode on one of the blebs formed at the cut ends of axons during slice preparation. Somatic current injections were used to trigger APs and the time between somatic and the axonal AP was measured (*Figure 2A*, *Figure 2—figure supplement 1C*). We captured two-photon images during electrophysiological recording and measured the length of the axonal path from the somatic to the axonal electrode on image z-stacks. The dataset was restricted to recordings that matched the distances from the soma to axodendritic close appositions determined above along the axon of synaptically coupled PC-to-PC connections (rat: n=8, 268.203±76.149 µm vs. human: n=9, 281.507±125.681 µm, two-sample *t*-test: p=0.799, *Figure 2F*). The latency between the soma and the axon bleb of the propagating AP peaks was not significantly different between the species (rat: n=8, 0.333±0.211 ms vs. human: n=9, 0.327±0.123 ms, two-sample *t*-test: p=0.945). The axonal speed of AP propagation was calculated for each cell from the time required from soma to recording site. We found no significant difference in the propagation speed of APs in the axons of rats and humans (rat: n=8, 0.848±0.291 m/s vs. human: n=9, 0.851±0.387 m/s, two-sample *t*-test: p=0.282, *Figure 2F*). Our axonal recordings suggest that there is no significant difference between the two species over the range of distances we investigated, so the lower latencies in the paired recordings may be due to dendritic differences.

We next sought to test rat and human dendritic signal propagation velocity using simultaneous whole cell patch clamp recordings with electrodes placed on the somata and dendritic shafts of PCs. Distances of somatic and dendritic recording locations (rat: 143.078±72.422 µm, n=46; vs. human: 153.446±57.698 µm, n=62, Mann-Whitney test: p=0.175, *Figure 2B*) were chosen to be similar in the

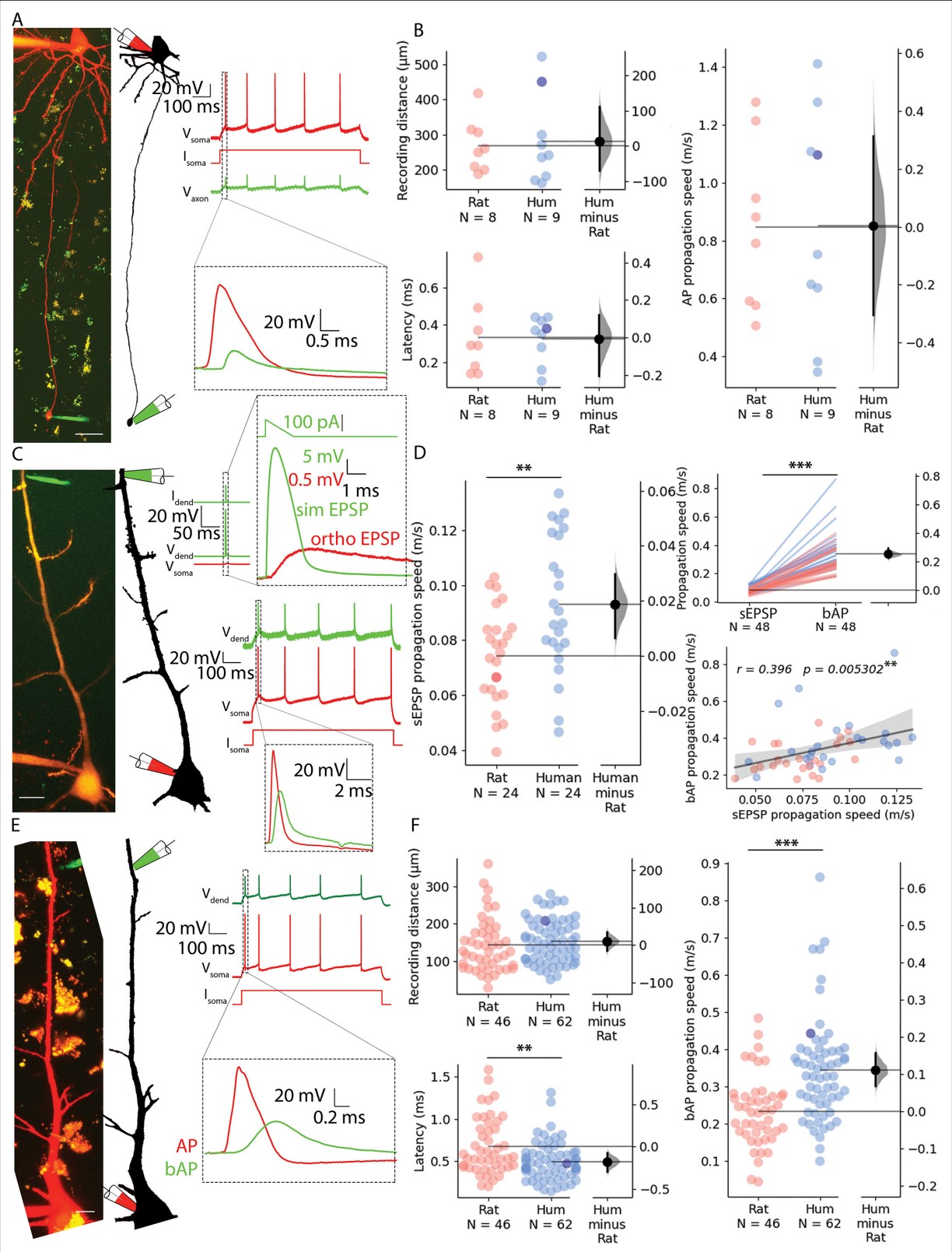

**Figure 2.** Propagation velocity of dendritic and axonal signals in rat and human cortical pyramidal cells. (**A**) Left, Human pyramidal cell simultaneously recorded with a somatic (red pipette) and axonal (green pipette) electrode. Right, Somatic depolarizing current ($I_{soma}$) evoked action potentials ($V_{soma}$) and their propagation to the axonal recording site ($V_{axon}$). (**B**) Path distances and AP latencies measured between the soma and axon bleb. AP propagation speed measured along the axon showed no significant difference (two-sample t-test: p=0.986). All recordings were made at resting membrane potential.

*Figure 2 continued on next page*

*Figure 2 continued*

(**C**) Left, Two-photon image of a rat pyramidal cell recorded simultaneously with a somatic (red pipette) and dendritic (green pipette) electrode. Top, Dendritic stimulation ($I_{dend}$) with simulated excitatory postsynaptic potentials (EPSP) waveform ($V_{dend}$) and somatic response ($V_{soma}$). Bottom, Somatic stimulation ($I_{soma}$) triggers an AP ($V_{soma}$) detected in the dendrite as bAP ($V_{dend}$). (**D**) Left, simulated EPSP propagation speed in rat and human cells (rat: 0.074±0.018 m/s vs. human: 0.093±0.025 m/s, two-sample t-test: p=0.004). Top right, simulated EPSP dendritic propagation speed was lower than bAP propagation speed (sEPSP: 0.084±0.023 m/s vs. bAP: 0.337±0.128 m/s, Wilcoxon signed ranks test: p=1.631 × 10$^{-9}$). Bottom right: there was a significant correlation in the forward propagating sEPSP speed and the speed of bAPs. Darker dot is the data for the cell shown on panel **C**. (**E**) Left, Two-photon image and reconstruction of a human pyramidal cell recorded simultaneously with a somatic (red pipette) and dendritic (green pipette) electrode. Right, Somatic current ($I_{soma}$) evoked APs ($V_{soma}$) and their backpropagation into the dendritic recording site ($V_{dend}$). (**F**) Top left, recording distance. Lower left, bAP latency was shorter in human cells (Mann-Whitney test: p=0.005). Right, bAP propagation speed was significantly higher in human dendrites (Mann-Whitney test: p=6.369 × 10$^{-6}$). Darker dot indicate the data for the cell shown on panel E. Scale bars A and C: 10 µm, E: 20 µm. *p<0.05, **p<0.01, ***p<0.001.

The online version of this article includes the following figure supplement(s) for figure 2:

**Figure supplement 1.** Latencies and propagation speed measured at different points of the propagating waveforms.

two species and in range of soma-to-dendrite distances of axo-dendritic close appositions determined above for synaptically coupled PC-to-PC connections. In the first set of experiments, we injected suprathreshold current through the somatic electrode and measured the time difference between the evoked AP peak at the soma and the respective backpropagating AP peak in the dendritic electrode (*Figure 2E and F*). We found significant difference in the signal propagation time between rat and human PCs (rat: 0.672±0.334 ms, n=46; vs. human: 0.495±0.229 ms, n=62, Mann-Whitney test: p=0.005, *Figure 2F*). The AP propagation speed was calculated for each cell from the time difference between the somatic and dendritic APs divided by the distance between the two points. We found that the propagation speed was, on average, ~1.47 fold faster in human (rat: 0.233±0.095 m/s vs. human: 0.344±0.139 m/s, Mann-Whitney test: p=6.369 × 10$^{-6}$, *Figure 2F*, *Figure 2—figure supplement 1B*). In a second set of experiments, using the same dual recording configuration, we tested orthodromic or forward propagating signal propagation velocity by injecting short-duration current ramps to simulate EPSP (sEPSP) signals in the dendrites and recorded the resultant subthreshold voltage response in the soma (*Figure 2C*). These experiments were performed in the same PCs where backpropagating AP velocities were also measured (rat: n=24, human: n=24). We found that sEPSP propagation speed was, on average, ~1.26 fold faster in human (rat: 0.074±0.018 m/s vs. human: 0.093±0.025 m/s, two-sample *t*-test: p=0.004; *Figure 2D*, *Figure 2—figure supplement 1D*). In addition, we found correlation between forward propagating sEPSP speed and back propagating AP speed (Pearson correlation coefficient, *r*=0.396, p=0.0053, *Figure 2D*).

## Contribution of ion channels of the dendritic membrane to signal propagation velocity

Hyperpolarization-activated cyclic nucleotide-modulated (HCN) channel densities were shown to be higher in human compared to rat layer 2/3 PCs and were shown to be instrumental in more depolarized resting membrane potentials and in larger sag potentials in response to hyperpolarization in the human (*Kalmbach et al., 2018*). In addition, modeling predicted that signal delay in dendrites reduces with increased h-conductance (*Kalmbach et al., 2018*). In line with previous studies, human PCs in our dataset had more depolarized resting membrane potential (rat: –70.49±5.78 mV, human: –64.30±7.28 mV, Mann-Whitney U test: p=7.37 × 10$^{-6}$, *Figure 3—figure supplement 1A*) but the average somatic input resistance were not significantly different in the two species (rat: 59.56±21.86 MΩ, n=46, human: 71.375±65.485 MΩ, n=62, Mann-Whitney test: p=0.347, *Figure 3—figure supplement 1A*).

Based on the correlation found between forward-propagating sEPSP speed and back-propagating AP speed, we performed pharmacological experiments on bAPs (since it is technically less challenging to evoke) to uncover potential contributors to increased dendritic speeds in humans. To test the contribution of h-channels to the elevated signal propagation speed in human dendrites, we performed pharmacological experiments with 20 µM ZD7288, a specific blocker of h-channels. Significant hyperpolarization of the resting membrane potential was observed in the human cells but not in the rat neurons (*Figure 3—figure supplement 1B*) and significantly increased input resistance accompanied drug application in both human and rat neurons (*Figure 3—figure supplement*

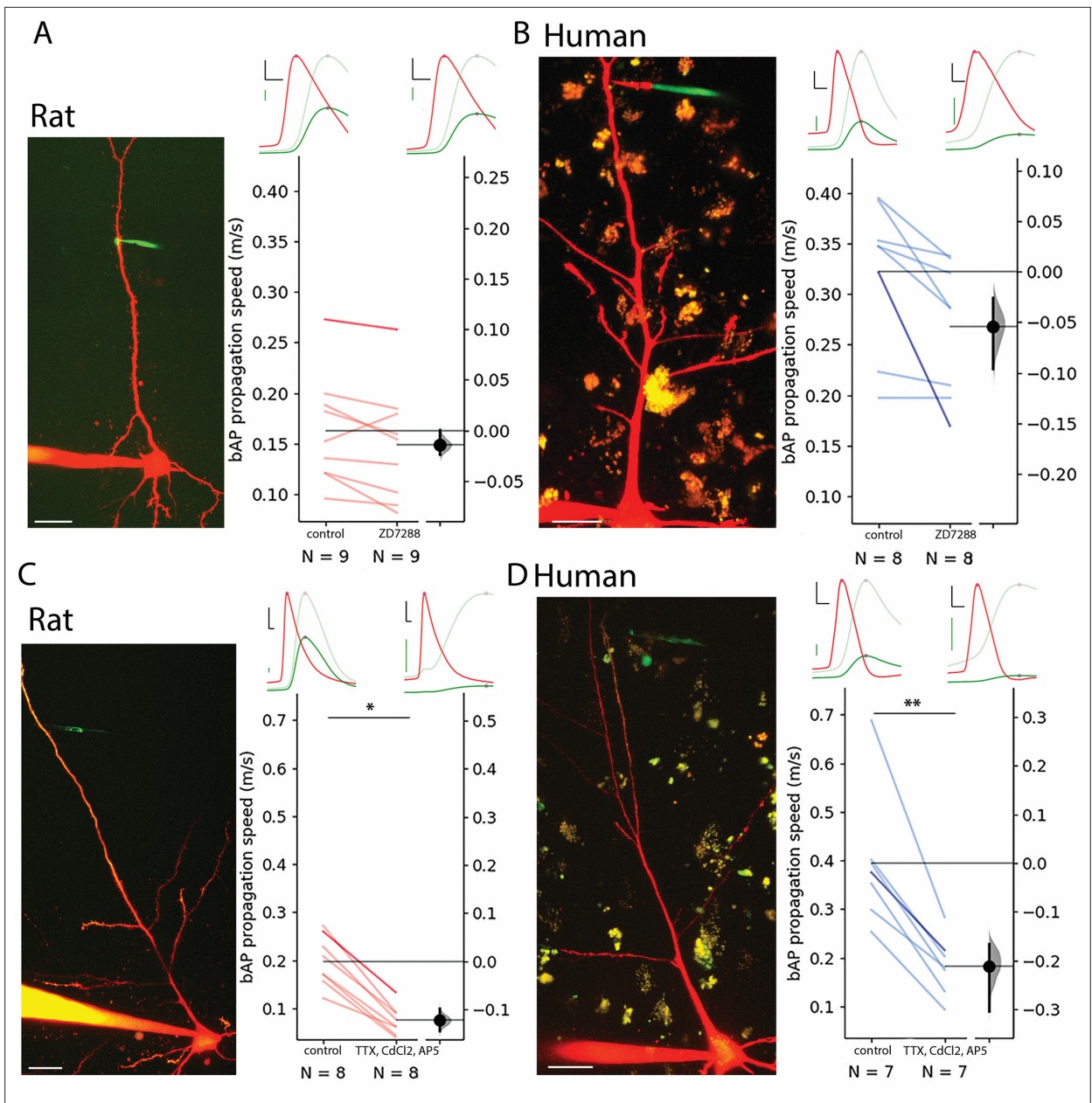

**Figure 3.** Contribution of HCN, $Ca^{2+}$, $Na^+$, and NMDA channels to bAP propagation speed in rat and human dendrites. (**A**) Representative recording from layer 2/3 pyramidal cell of a rat. Two-photon maximum intensity projection image of Alexa 594 and biocytin-filled neuron on the left, representative somatic action potential (AP) (red) and dendritic bAP (green) on the upper right in the control condition (left) and after 20 µM ZD7288 application (right). The light green represents the dendritic signal scaled to the amplitude of the somatic signal for better visibility. Effect of ZD7288 on bAP propagation speed. Darker color represents the example cell. (**B**) Same as in panel A but for human cells. (**C and D**) Same as **A** and **B** but the ACSF contained 1 µM TTX, 200 µM $CdCl_2$, and 20 µM AP5 in the drug application condition. Black scale bars 20 mV and 0.3 ms. Green scale bars 5 mV on **A** and **B**, 2.5 mV on **C** and **D**. Scale bars on microphotographs 20 µm. All recordings were done on resting membrane potential. *p<0.05, **p<0.01.

The online version of this article includes the following figure supplement(s) for figure 3:

**Figure supplement 1.** Properties of dendro-somatic recording and measured membrane parameters.

*1C*). Drug administration did not significantly decrease bAP propagation speed in rat PCs (control: 0.163±0.054 m/s, ZD7288: 0.149±0.057 m/s, n=9, two-way ANOVA with repeated measures and Tukey HSD (Honestly Significant Difference) post-hoc comparison: p=0.9, *Figure 3A*, *Figure 2— figure supplement 1E*) and in human PCs (control: 0.322±0.073 m/s, ZD7288: 0.268±0.066 m/s, n=8, two-way ANOVA with repeated measures and Tukey HSD post-hoc comparison: p=0.329, *Figure 3B*,

*Figure 2—figure supplement 1E*). The human dendrites maintained a higher bAP propagation speed in response to h-channel blockage (two-way ANOVA with repeated measures and Tukey HSD post-hoc comparison: p=0.003). We could not find significant interaction of species and ZD7288 treatment (two-way ANOVA with repeated measures: interaction p=0.358). These results suggest that the species-dependent density difference of HCN channels of pyramidal apical dendrites do not by themselves explain the propagation speed differences between the two species.

Back-propagation of APs is an active process supported by voltage-gated ion channels that can initiate regenerative events in the dendrites (*Stuart and Sakmann, 1994*). To further investigate the influence of voltage-gated ion channels we pharmacologically blocked voltage-gated $Na^+$ channels with tetrodotoxin (TTX, 1 μM), voltage-gated $Ca^{2+}$ channels with cadmium chloride ($CdCl_2$, 200 μM), and NMDA receptors with (2 R)-amino-5-phosphonovaleric acid (AP5, 20 μM) simultaneously. Since the blockage of voltage-gated $Na^+$ channels prevent the initiation of APs, we kept the soma of the recorded cells in voltage clamp mode and used a prerecorded template as voltage command through a somatically placed electrode (the so called 'simulated spike') and measured the back propagation of the response to the somatic voltage command at a dendritic recording site in current clamp mode. As expected, the amplitude of the bAPs at the dendritic recording site dropped significantly in human and rat cells, respectively (*Figure 3—figure supplement 1D*). The speed of back propagation of membrane potential signals in dendrites with blocked regenerative events by the pharmacological cocktail was significantly reduced in rat samples compared to the drug-free control (rat control: 0.199±0.053 m/s, rat TTX/$CdCl_2$/AP5: 0.076±0.03 m/s, two-way ANOVA and Tukey HSD post-hoc comparison: p=0.024, *Figure 3—figure supplement 1D*, *Figure 2—figure supplement 1F*), and was significantly lower in human samples as well (human control: 0.395±0.14 m/s, human TTX/$CdCl_2$/AP5: 0.184±0.061 m/s, two-way ANOVA and Tukey HSD post-hoc comparison: p=0.001, *Figure 3E*, *Figure 2—figure supplement 1F*). The human dendrites with blocked action potential generation on average had a higher bAP propagation speed, however, it was not statistically significant (rat: 0.076±0.03 m/s n=8, human: 0.184±0.061 m/s n=8, two-way ANOVA with repeated measures and Tukey HSD post-hoc comparison: p=0.066). We could not find significant interaction of species and drug treatment (two-way ANOVA with repeated measures, interaction: p=0.142). In summary, in the search for factors contributing to higher signal propagation speeds in human pyramidal dendrites compared to rat pyramidal dendrites, ion channels such as voltage-gated $Na^+$, $Ca^{2+}$, and NMDA channels and the HCN seem to play a minor role in distinguishing the two species.

## Specific membrane capacitance

The specific membrane capacitance ($C_m$) can influence the time constant of the biological membrane, and it is a key determinant of the propagation of electrical signals. Recent experiments indicated that the $C_m$ of human L2/3 PCs might be significantly lower compared to rodents (*Eyal et al., 2016*) and modeling studies suggested that the decrease in $C_m$ could lead to increased conduction speed and fewer synapses being able to evoke suprathreshold events in human PCs (*Eyal et al., 2016*). However, a separate line of experiments could not detect differences in the $C_m$ of L5 PCs between humans and rodents (*Beaulieu-Laroche et al., 2018*), or L2/3 PCs (*Gooch et al., 2022*) thus, to test whether $C_m$ is a component in producing elevated signal propagation velocity in human dendrites, we directly measured the $C_m$ values of human and rat PCs by pulling nucleated patches (*Eyal et al., 2016*; *Figure 4A and B*). We found no significant difference in the $C_m$ between the human and rat L2/3 PCs (rat: 1.092±0.14 μF/cm$^2$ n=20; human: 0.987±0.196 μF/cm$^2$ n=19, two-sample *t*-test: p=0.0615, *Figure 4C*). The specific membrane capacitance is determined by the dielectric constant of the membrane, and it is inversely proportional with the membrane thickness. We measured the membrane thickness of dendritic structures with transmission electron microscopy both in human and rat samples (*Figure 4D and E*) and detected no significant differences between the two species (human: 4.271±0.873 nm, n=213 from n=3 patient; rat: 4.122±0.779 nm n=151 from n=3 rat, Mann-Whitney test: p=0.212, *Figure 4E*). Based on these experiments it seems that not the specific membrane capacitance is the key determinant of the higher signal propagation speed in human cells.

## Effect of dendritic thickness

The relationship between conduction velocity and axon diameter is well known for small myelinated and unmyelinated axons (*Waxman and Bennett, 1972*). Anatomical features of neuronal dendrites

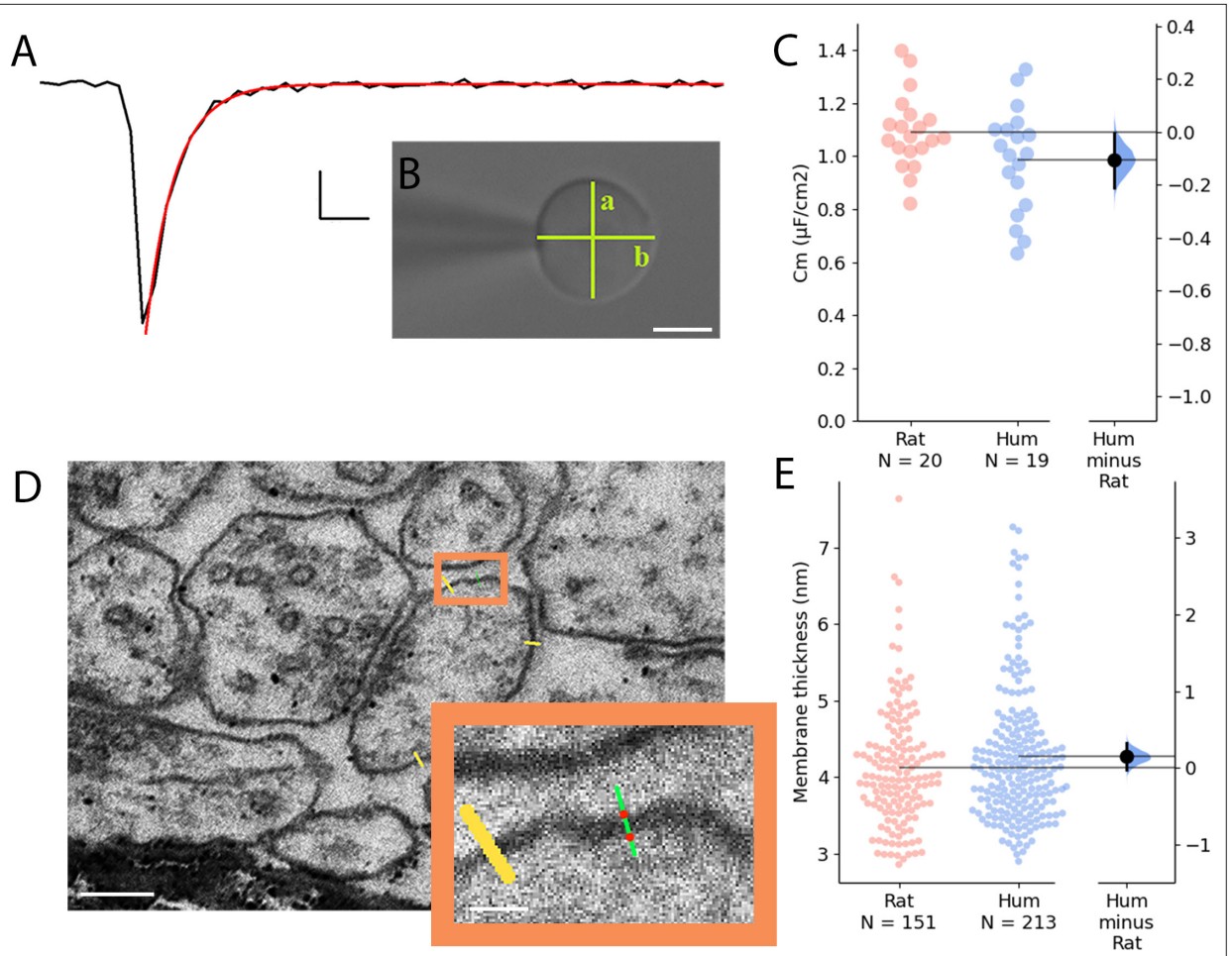

**Figure 4.** Comparative analysis of membrane capacitance and thickness in rat and human cortex. (**A**) Representative capacitive transient of a nucleated patch pulled from layer 2/3 neocortical pyramidal cell (black). A single exponential function was fitted on the measured signal (red) for the calculation of the time constant of the membrane. Scale bar: 100 pA, 20 µs. (**B**) Differential interference contrast microscope image of a neuronal nucleus. The shortest (**a**) and longest (**b**) diameter values were used to calculate the membrane surface. Scale bar 5 µm. (**C**) Specific membrane capacitance of rat (red) and human (blue) neocortical pyramidal cells. (**D**) Electron micrographs of dendritic membranes used for membrane thickness measurements. Yellow lines indicate measuring profiles. Scale bar 40 nm. Boxed region magnified on the right. The two red dots on the green line show the edges of the membrane (see methods). Inset scale bar 10 nm. (**E**) Membrane thickness of rat (red, n=151 from n=3 rat) and human (blue, n=213 from n=3 patient) neocortical cell dendrites (Mann-Whitney test: p=0.212).

also have a major influence on signal propagation properties (*Deitcher et al., 2017*; *Rall and Rinzel, 1973*; *Rinzel and Rall, 1974*; *Vetter et al., 2001*), thus, in addition to the soma-dendritic path measurements shown above, we also measured the thickness of dendrites at every 0.5 µm along the path linking the somatic and dendritic electrodes on two-photon image stacks captured during electrophysiological measurements (*Figure 5A–C*). We found that the mean diameter of dendrites was thicker in human (2.272±0.584 µm, n=62) compared to the rat (2.032±0.413 µm, n=46, two-sample *t*-test: p=0.019, *Figure 5D*). Moreover, in samples where we acquired both dendrite thickness and bAP signal propagation velocity, we found that the mean dendritic diameter between the recording sites was correlated with the speed of backpropagating APs (*Figure 5E*).

## Modeling EPSP propagation in dendrites

Detailed compartmental models were utilized to disassemble the effect of various morphological and cable parameters on the latency and velocity of synaptic potential in human and rat L2/3 dendrites. Based on the 3D morphological reconstructions of five human and four rat PCs (*Figure 6—figure supplement 1*), we first asked, how dendritic morphological differences per se affect signal

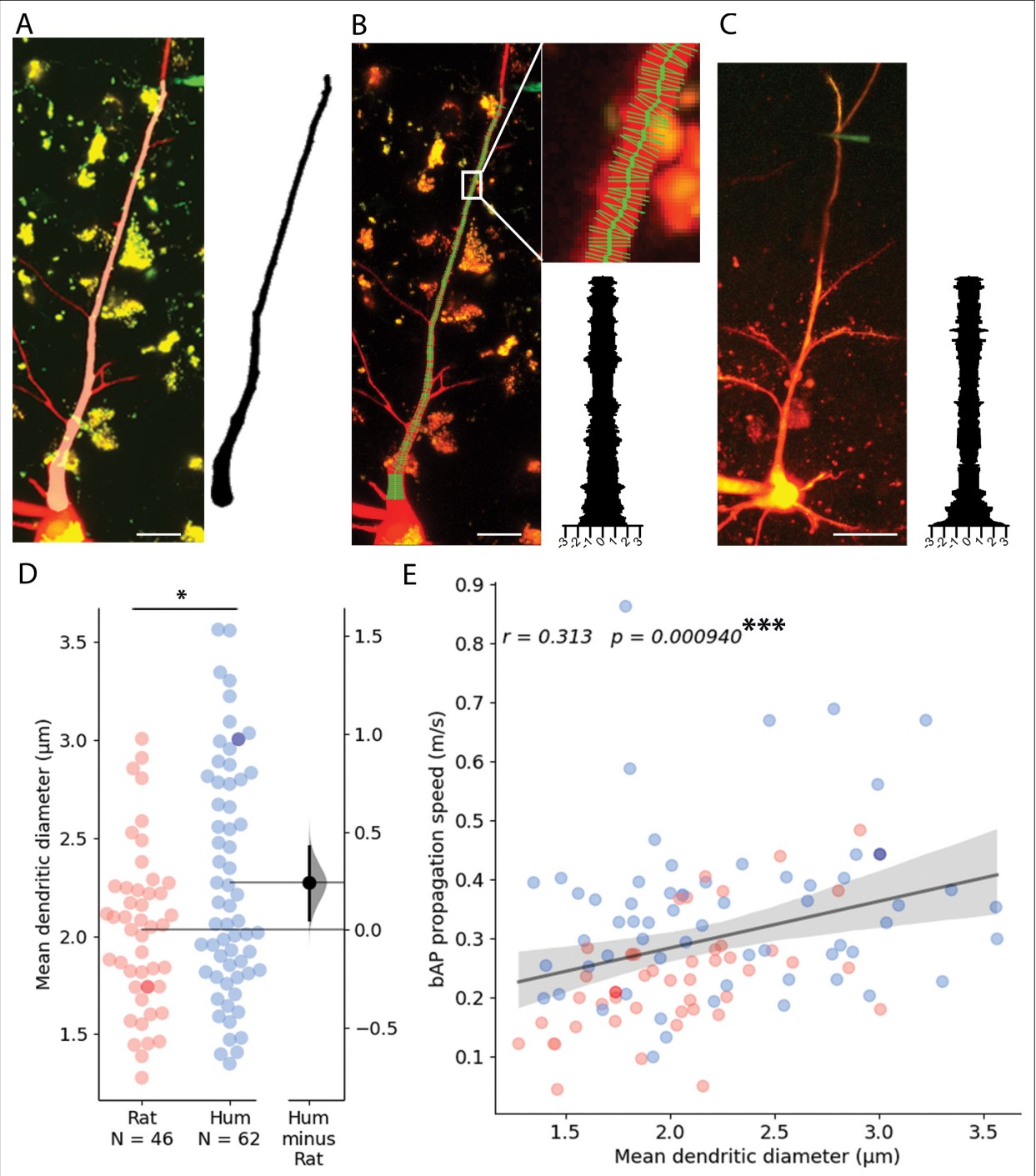

**Figure 5.** Dendritic thickness reconstructions and comparison of layer 2/3 pyramidal cells in the human and rat cortex. (**A**) Left, Maximum intensity projection of Alexa594 and biocytin-filled human pyramidal cell imaged in two-photon microscope. Right, model of 3D reconstructed apical dendrite. Middle, overlay of the two-photon image and the model. (**B**) Apical dendrite thickness measurements on the sample shown in **A**. Left, The center of the dendrite is tracked by a thick green line while the perpendicular thin lines show measured profiles. Right, Stacked thickness measurements with micrometer scale. (**C**) Same as in B with a rat sample. Scale bars 20 µm. (**D**) Comparison of rat and human apical dendrite averaged thickness. The mean dendritic diameter of human dendrites was significantly thicker than rat ones (two-sample *t*-test: p=0.019). Darker dots indicate data measured on image stacks shown in panel **B** and **C**. (**E**) bAP propagation speed correlates significantly with dendrite thickness. Pearson correlation coefficient (**r**) values for fitted lines are shown on the upper left corner of the plot. The shaded area around the regression line shows the 0–100% confidence interval for the bootstrapped data. ***p<0.001.

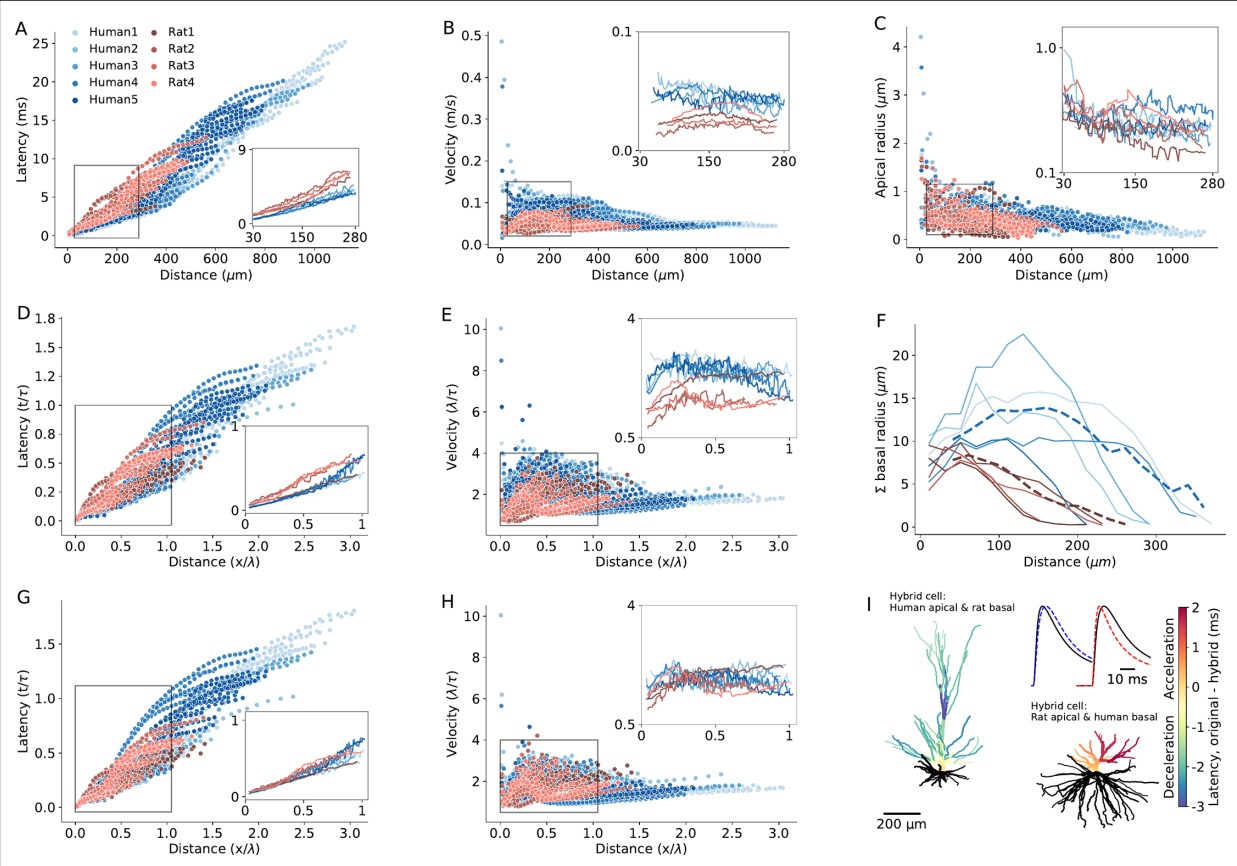

**Figure 6.** Modeling explains the enhanced excitatory postsynaptic potentials (EPSPs) velocity in the apical dendrites of human L2/3 PCs. (**A**) Latency and (**B**) velocity of EPSP in models of 5 human (blue) and 4 rat (red) reconstructed L2/3 PCs. Insets show the respective averages for the zoom-in region (box), which brackets the experimental range of dendritic recordings. Note the smaller latency and larger velocity in human PCs. (**C**) Dendritic radius as a function of distance from the soma. Note the larger radius of human dendrites in the outlined region. (**D and E**) As A and B, but now distance is normalized in cable units (thus incorporating the diameters differences between cells) and time is normalized in units of membrane time constant. (**F**) Sum of radii of basal dendrites as a function of distance from the soma (blue – human, red – rat), in 20 µm bins. Dashed lines are the respective averages. (**G and H**) As D and E but for 'hybrid cells', computed for the five modeled human neurons, all having the basal tree of 'Rat4' (blue) and for the four modeled rat cells, all with the basal tree of 'Rat4' (red). Note that the differences in latency and velocity between human and rat diminished (insets). (**I**) Two examples of a color-coded 'latency-gram,' visualizing the effect of replacing the basal tree of human L2/3 PC with the basal tree of rat L2/3 PC and *vice versa*. Top left: 'Human1' apical tree with basal tree (in black) of 'Rat4' PC. Lower left: 'Rat4' apical tree with basal tree of 'Human1.' Color-coded difference in latency was calculated by subtracting the respective values of the original cells from those calculated for the 'hybrid cells.' The blueish apical tree of the human apical tree indicates deceleration whereas the reddish apical tree of the rat PC indicates acceleration of the EPSPs. Inset shows examples of a somatic EPSP's in these two cases. Shown are the original EPSPs (black lines) and the EPSPs computed for the respective hybrid cases (dashed line in blue – deceleration; dashed red line – acceleration) both for synaptic inputs at 288 µm from soma. Specific cable properties in all cells were: $C_m$=1 µF/cm$^2$, $R_m$=15,000 Ωcm$^2$, $R_a$=150 Ωcm.

The online version of this article includes the following figure supplement(s) for figure 6:

**Figure supplement 1.** Morphology of the nine modeled cells.

**Figure supplement 2.** 'Hybrid cells' effect on latency and velocity for the experimentally-uniform vs fitted cable parameters.

**Figure supplement 3.** Basal load effect.

**Figure supplement 4.** Quantifying the effect of switching the basal tree between rat and human and vice versa- the 'hybrid cells' on mean latency, uniform parameters.

propagation, assuming that the cable parameters are identical in all cells ($C_m$=1 *µF/cm$^2$*, $R_m$=15,000 *Ωcm$^2$*, $R_a$=150 *Ωcm*, *Figure 6*). *Figure 6A and B* shows EPSPs latency and velocity as a function of distance from its dendritic initiation site to the soma. Latency was calculated as the time difference between the peak-times of the local dendritic EPSP and of the resulting somatic EPSP. The dendritic-to-soma latency ranged between 0.1–13 ms in rats (red circles) and 0.01–25 ms in human (blue circles). The larger maximal latency in human is expected due to the ~twofolds longer apical dendrite in

humans L2/3 neurons (*Figure 6A*). The respective EPSPs velocity was calculated by dividing the path distance to the soma from the dendritic origin of the EPSP by its latency (*Figure 6B*). EPSP velocity ranged between 0.02–0.09 m/s in rat and 0.01–0.48 m/s in human (*Figure 6B*). The exceptionally large differences in the maximal velocity between human and rat is taken in the Discussion.

We next compared signal propagation in rat and human dendrites, focusing on identical range of dendrite-to-soma distances in which the experiments were performed (27 μm – 289 μm, insets in *Figure 6A and B*). Towards this end, we computed the mean EPSP latency and velocity as a function of distance from the soma, averaged across different branches at a given distance from the soma (*Figure 6A and B*, lower right and upper right insets). For this experimental range of recordings, EPSP velocity ranged between 0.04–0.08 m/s in humans versus 0.03–0.05 m/s in rats. EPSP latency ranged between 0.1–4.7 ms in humans versus 0.4–6.5 ms in rats. These findings demonstrate significantly faster EPSP propagation in humans compared to rats (average latency in humans: 2.45±0.2 ms, n=5; in rats: 3.3±0.3 ms, n=4; Mann-Whitney U test: p=0.03. Average velocity in humans: 0.08±0.005 m/s, n=5; in rats: 0.05±0.006 m/s, n=4; Mann-Whitney U-test: p=0.02. See *Figure 6—figure supplement 2* and *Supplementary file 2*).

A possible reason for the smaller latency and larger velocity of EPSPs in human apical dendrites is that they have larger diameter (*Figures 5D and 6C* see also *Agmon-Snir and Segev, 1993*; *Jack et al., 1975*). Theory shows that, for an infinitely long passive cylindrical cable, the velocity of passive signals is not constant. It is fast near their site of origin, converging to a value of $2\lambda/\tau$ away from their initiation site (*Agmon-Snir and Segev, 1993*; *Jack et al., 1975*) ($\lambda$ is the cables' space constant and $\tau$ is its membrane time constant). This means that the latency and velocity of passive signals, when normalized in units of $\lambda/\tau$, are identical for cylindrical cables with different diameters. This is due to the fact that differences in cable diameter are taken into account when normalizing the physical distance, x, by $\lambda$ (which is $\propto \sqrt{d}$, where *d* is the cable diameter). Hence, if the larger diameter in human dendrites is a key contributor to the enhanced signal velocity in these cells, we expect that the EPSP latency and velocity will converge on similar curves for all cells (rat and human alike) after normalizing the distance in units of $\lambda$, and time in units of $\tau$. However, albeit such normalization, the velocity is still larger and the latency is shorter in human (compare insets in *Figure 6D and E* to *Figure 6A and B*, respectively. In this case, the average latency in humans: 0.16±0.01 $\tau$, n=5; in rats: 0.22±0.02 $\tau$, n=4; Mann-Whitney U test: p=0.02. The average velocity in humans: 2.55±0.11 $\lambda/\tau$, n=5; in rats: 1.73±0.29 $\lambda/\tau$, n=4; Mann-Whitney U test: p=0.02).

To summarize: Scaling dendritic distance in units of $\lambda$ and time in units of $\tau$ did not eliminate the statistically significant differences in EPSP latency and velocity between humans and rats. This raises the question: what factor enhances EPSP propagation speed in human dendrites?

One possibility is that differences in boundary conditions for EPSPs traveling from the dendrites towards the soma might explain the enhanced signal propagation in human. Boundary conditions are known to affect the steepness of voltage attenuation along the dendrites (*Rall and Rinzel, 1973*; *Rinzel and Rall, 1974*). But do differences in the boundary condition ('the impedance load') at the soma affect the speed of EPSPs propagating when traveling from the apical tree towards the soma? Notably, the basal tree in human L2/3 PCs is substantially larger than that of rat (*Figure 6F* and 8A). Quantifying the total membrane area confirms that the basal tree in human is significantly larger than in rats (in humans: 23,621±7735 μm², n=5; in rats: 9127±2759 μm², n=4; Mann-Whitney U test: p=0.02, see *Figure 6—figure supplement 3*). Consequently, a larger impedance load (larger 'sink') is expected at the soma in human L2/3 neurons.

To examine the impact of impedance load, we computationally substituted the basal tree of human neurons with the basal tree of rat and *vice versa* (creating 'hybrid cells'). This substitution diminished the inter-species differences in latency and velocity (both in the original units and after normalizing the distance and time in units of $\lambda$ and $\tau$, respectively). Examples of these 'hybrid cells' are shown in *Figure 6G and H*. In these cases, the basal trees of the 5 modeled human neurons (blue dots) and the basal tree of 'Rat1,' 'Rat2,' and 'Rat3' (red dots) were all replaced with the basal tree of 'Rat4' neuron. This resulted in a significant reduction in EPSP velocity in the human neurons, diminishing the differences in signal latency/velocity between humans and rats (average latency in humans: 3.42±0.5 ms or 0.23±0.03 in units of $\tau$, n=5; in rats: 3.2±0.2 ms, 0.21±0.01 in units of $\tau$, n=4; Mann-Whitney U test: p=0.7 for both ms and $\tau$ units. Average velocity in humans: 0.06±0.005 m/s or 1.86±0.1 $\lambda/\tau$, n=5; in rats: 0.05±0.005 m/s or 1.8±0.1 $\lambda/\tau$, n=4; Mann-Whitney U test: p=0.73 for m/s units and p=0.66

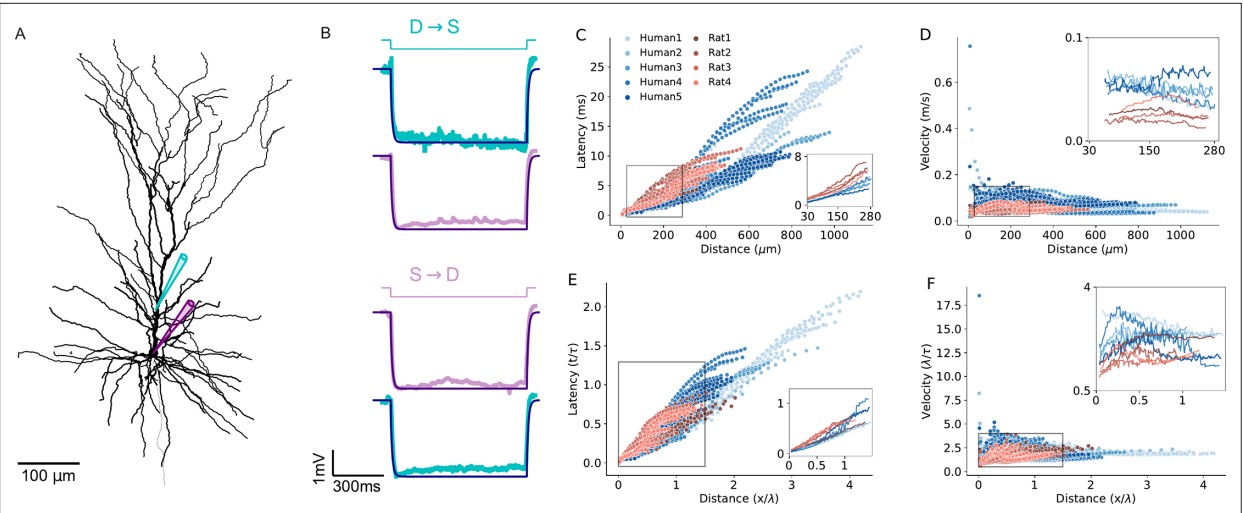

**Figure 7.** Modeling excitatory postsynaptic potentials (EPSPs) latency and velocity in dendrites of human and rat L2/3 pyramidal cells based on experimentally-fitted cable parameters. (**A**) Exemplar modeled ('Human5') L2/3 PC, also showing the locations of the two recording/stimulating electrodes. (**B**) Top (D→S): step hyperpolarizing current (–100 pA) injected to the dendrite of the modeled cell (cyan). Lower trace: Model fit (dark purple line) to the voltage response at the soma (noisy light purple line). The resultant fit to the local dendritic voltage response is also shown (in cyan). Bottom (S→D): as is the case at top, but with current step injected to the soma (purple step current). This fitting procedure resulted with the following passive parameters: $C_m$=0.63 $\mu F/cm^2$, $R_m$=15,570 $\Omega cm^2$, $R_a$=109 $\Omega cm$. (**C**) Latency and (**D**) velocity of EPSPs for the 9 model cells as in **Figure 6A and B**, but now with specific cable parameters fitted to the individual modeled neurons (see **Table 1**). (**E and F**) As in **C** and **D**, with distance normalized in cable units and time normalized by the membrane time constant (see **Table 2**). Note the smaller latency and larger velocity for the human PCs, which is now more significant as compared to the case where the cable parameters were uniform for all modeled cells (compare to **Figure 6D and E**).

The online version of this article includes the following figure supplement(s) for figure 7:

**Figure supplement 1.** Morphological irregularities affect excitatory postsynaptic potentials (EPSP) latency and velocity.

**Figure supplement 2.** Passive cable parameters fitted to experimental data.

**Figure supplement 3.** 'Hybrid cells' effect on latency and velocity for the experimentally-fitted cable parameters.

for $\lambda / \tau$ units. See **Figure 6—figure supplement 2** and **Supplementary file 3**). Repeating this procedure for all modeled PCs, but now the basal of any given cell (of both human and rat) was replaced, one-by-one, by the basal tree of all other cells. Again, this confirmed that the significant difference in EPSPs latency between humans and rats consistently diminished and became insignificant in the 'hybrid cell' manipulations. More specifically, human basal trees on rat cells accelerate the EPSPs, whereas rat basal trees on human neurons decelerate the EPSPs (**Figure 6—figure supplement 4**).

To further demonstrate the effect of switching the basal tree between human and rat neurons ('hybrid cells') on the EPSPs' velocity and latency, we depict in **Figure 6I** the case where the basal tree of 'Human1' PC was replaced with the basal tree of 'Rat4' (top left) and *vice versa* (lower right). The result (the 'latency-gram' of the EPSPs) is depicted in color-code, showing deceleration in the apical tree of the human cell (top left) and acceleration in the rat's apical tree (lower right) due to these manipulations. Exemplar somatic EPSPs originated at ~282 μm from the soma in the original cells (black line) and in the respective 'hybrid case' (dashed lines) are shown at top right. The deceleration in the case of a human cell with a rat basal tree is shown on the left and the acceleration in the case of a rat cell with a human basal tree is shown on the right. The explanation for the surprising large impact of the impedance load of the basal tree on signal propagation in the apical tree is elaborated in the Discussion.

In addition to morphological features influencing EPSP latency and velocity (**Figure 7**, **Figure 7—figure supplement 1**), the three key passive parameters - specific membrane resistivity ($R_m$), capacitance ($C_m$) and axial resistivity ($R_a$) - affect signal propagation properties in dendrites by altering the cables' space constant ($\lambda$) and membrane time constant ($\tau$). These changes can either enhance or compensate for the increased velocity of signal propagation of human cells, depending on the specific values of these parameters. Thus, we next examined to what extent the cable parameters of the individual PCs studied here influence signal propagation in their respective dendrites. To address

**Table 1.** Passive cable parameters fitted to experimental data.

$C_m$ and $R_m$ are the specific membrane capacitance and resistivity, respectively; $R_a$ is the specific axial resistance.

| Cell name | $C_m$ (µF/cm²) | $R_m$ (Ωcm²) | $R_a$ (Ωcm) |
| --- | --- | --- | --- |
| Human1 | 0.65 | 19,875 | 298 |
| Human2 | 0.60 | 15,672 | 263 |
| Human3 | 0.85 | 12,872 | 103 |
| Human4 | 0.77 | 21,523 | 209 |
| Human5 | 0.63 | 15,570 | 109 |
| Rat1 | 0.84 | 13,110 | 267 |
| Rat2 | 1.16 | 9084 | 249 |
| Rat3 | 1.02 | 14,497 | 115 |
| Rat4 | 1.41 | 8527 | 109 |
| Mean human | 0.70 | 17,120 | 196 |
| Mean rat | 1.11 | 11,304 | 185 |

this, we fitted the cable parameters for each of the 9 reconstructed PCs individually, based on double-electrodes recordings (soma and dendrite) for each cell. *Figure 7A* shows an exemplar reconstructed human L2/3 PC (Human5) with the locations of the two recording/stimulating electrodes used for this cell. *Figure 7B* top (D-S: dendrite-to-soma direction) shows the case where the step current was injected at the dendrite (cyan). The resultant voltage response is depicted in cyan in the trace below; the model fit is superimposed in dark blue. The opposite (S-to-D) direction is depicted by the next three traces below. This fit enabled a direct estimate of the cable parameters per cell (*Table 1*). We found that $R_m$ is larger in humans and $C_m$ is smaller in humans (*Table 1*). Yet the membrane time constant ($\tau = R_m * C_m$) is statistically similar in the two species (*Table 2* and see *Figure 7—figure supplement 2*).

**Table 2.** Morphological and cable parameters, and model prediction, of the average excitatory postsynaptic potentials (EPSPs) latency and velocity within experimental range of dendritic recordings per modeled cell.

Cable parameters were fitted per cell as in *Table 1*. $l_{avg}$ , and $d_{avg}$- the average physical distance and diameter, respectively from which the respective experiments (per cell) were performed (zoom-in region in *Figure 7C and D*). $L_{avg}$ is the respective distances in cable units ($L = \frac{l}{\lambda}$); $\tau$ is the membrane time constant ($C_m * R_m$). Latency and velocity are the average values from dendrite to soma, computed for the experimental range of dendritic recordings.

| Cell name | $d_{avg}$ (µm) | $l_{avg}$ (µm) | $L_{avg}$($\lambda$ | $\tau$(ms) | Latency (ms) | Velocity (m/s) |
| --- | --- | --- | --- | --- | --- | --- |
| Human1 | 0.4 | 192 | 0.6 | 12.92 | 2.8 | 0.074 |
| Human2 | 0.5 | 180 | 0.54 | 9.46 | 2.18 | 0.089 |
| Human3 | 0.4 | 172.7 | 0.42 | 10.90 | 2.13 | 0.086 |
| Human4 | 0.6 | 178 | 0.41 | 16.60 | 2.5 | 0.074 |
| Human5 | 0.44 | 162.6 | 0.35 | 9.80 | 1.69 | 0.098 |
| Rat1 | 0.35 | 163.9 | 0.64 | 11.00 | 3.17 | 0.054 |
| Rat2 | 0.43 | 144.3 | 0.52 | 10.50 | 3.52 | 0.044 |
| Rat3 | 0.5 | 170.4 | 0.32 | 14.5 | 3.44 | 0.051 |
| Rat4 | 0.53 | 164.6 | 0.37 | 11.9 | 2.72 | 0.063 |
| Mean human | 0.47 | 177.1 | 0.46 | 11.9 | 2.26 | 0.085 |
| Mean rat | 0.45 | 160.8 | 0.46 | 12 | 3.21 | 0.053 |

*Figure 7C and F* extends the simulations using the fitted (rather than uniform) cable parameters for each cell (*Figure 6*). Compared to the uniform case, the differences in EPSP latency and propagation velocity between and within the two species are slightly enhanced (compare *Figure 7C and D*) to *Figure 6A and B*. For the per-cell fit, the latency ranges between 0.1–11 ms for rats (red) and 0.1–28 ms for humans (*Figure 7C*); the velocity ranges between 0.02–0.085 m/s for rats (red) and 0.02–0.75 m/s for humans (*Figure 7D*). After normalizing the distance by the space and time constants calculated per cell, the differences in both latency (*Figure 7E*) and velocity (*Figure 7F*) among individual cells is larger compared with the uniform case (*Figure 6D and E*). Importantly, despite this increased variance within species, the differences between humans and rats remain statistically significant, both with and without normalization (average latency in humans: 2.3±0.4 ms, or 0.19±0.03 $\tau$, n=5; in rats: 3.2±0.6 ms or 0.27±0.05 $\tau$, n=4; Mann-Whitney U test: p=0.03 for both ms and $\tau$ units. Average velocity in humans: 0.085±0.009 m/s or 2.5±0.4 $\lambda/\tau$, n=5; in rats: 0.05±0.0085 m/s or 1.7±0.3 $\lambda/\tau$, n=4; Mann-Whitney U test: p=0.03 for m/s and p=0.02 for $\lambda/\tau$ units. See *Figure 6—figure supplement 2* and *Table 2*).

Next, we applied our 'hybrid cells' method (as in *Figure 6G, H and I*). As a result, the inter-species differences were diminished (average latency in humans: 2.3±0.4 ms, 0.19±0.03 $\tau$, n=5; in rats: 3.2±0.6 ms, 0.27±0.05 $\tau$, n=4; Mann-Whitney U test: p=0.9 for ms units and p=1.0 $\tau$ units. Average velocity for human: 0.08±0.02 m/s, 2.39±0.2 $\lambda/\tau$, n=5; rat: 0.05±0.005 m/s, 1.7±0.4 $\lambda/\tau$ n=4; Mann-Whitney U test: p=0.2 for m/s and p=0.8 for $\lambda/\tau$ units. See *Figure 7—figure supplement 3*, *Figure 6—figure supplement 1* and *Supplementary file 4*).

To rank the impact of the various factors affecting EPSP propagation latency in human and rat neurons, we conducted a comprehensive statistical analysis using two complementary approaches: the generalized linear model (GLM) (*Kiebel and Holmes, 2007*) as well as SHAP (SHapley Additive exPlanations) (*Lundberg and Lee, 2017*) based on fitting Gradient Tree Boosting model (*Friedman, 2002*). We began by fitting a GLM without interaction terms among the factors affecting EPSP latency (*Supplementary file 5*). This enables us to quantify the primary individual factors affecting EPSP propagation. Our analysis revealed the following ranking order: (1) physical distance of synapses from soma had the strongest effect; (2) species differences; (3) conductance load, as demonstrated by our 'hybrid cells' manipulation; (4) radii of the apical dendrite, affecting the cables' space constant, $\lambda$; and (5) the specific cable parameters, as revealed when using per-cell fitted parameters versus uniform cable parameters, was minimal. We next performed GLM analysis with interaction terms showing that, as expected, there are significant interactions between the factors affecting EPSP

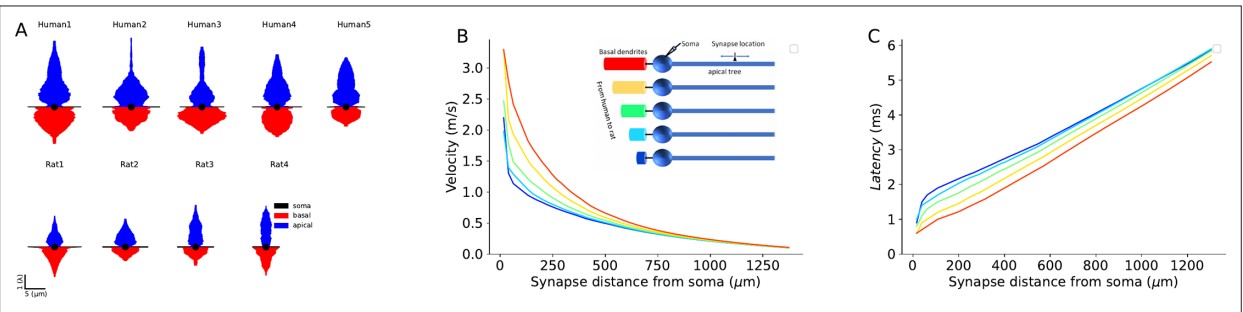

**Figure 8.** Impact of conductance load of the basal tree on excitatory postsynaptic potentials (EPSPs) velocity and latency. (**A**) Equivalent cable for the apical tree (in blue) and the basal tree (in red) for the 9 L2/3 cells modeled in this study. Note the relatively large conductance load (sink) imposed by the large basal tree in human cells. (**B**) EPSP velocity and (**C**) latency as a function of distance of the (apical) synapse from the soma. The synapse was located along the 'apical' cable (blue cylinder, inset). The respective five cases are shown in the inset. Velocity and latency were computed as in *Figures 6 and 7*. Note the enhanced velocity and reduced latency for larger basal dendrites. Cable parameters were: $C_m$=1 $\mu F/cm^2$, $R_m$=15,000 $\Omega cm^2$, $R_a$=150 $\Omega cm$. The apical cylinder is of infinite length with diameter of 3 $\mu m$; the basal tree (color cables) have linearly increasing diameter (**d**) and length (**L**), approximating the increment from rat to human basal trees (*Figure 6F*): red (l=800 $\mu m$, d=20 $\mu m$), yellow (l=700 $\mu m$, d=18 $\mu m$); green (l=600 $\mu m$, d=16 $\mu m$); light blue (l=500 $\mu m$, d=14 $\mu m$); dark blue (l=400 $\mu m$, d=12 $\mu m$). Soma diameter was 20 $\mu m$ in all cases.

The online version of this article includes the following figure supplement(s) for figure 8:

**Figure supplement 1.** SHapley Additive exPlanations (SHAP) analysis feature importance result.

**Figure supplement 2.** Effect of dendritic branching points on signal propagation velocity.

**Figure supplement 3.** Effect of series resistance of the dendritic electrode on measurement of excitatory postsynaptic potentials (EPSP) latency.

latency (*Supplementary file 6*). To further validate the above ranking while incorporating the interactions between the various factors affecting EPSP latency, we performed a SHAP analysis. Notably, even with interactions included, the ranking of the factors affecting signal propagation are aligned with the results from the analysis based on the GLM without interaction terms (see *Figure 8—figure supplement 1*).

We summarize this section by noting that our theoretical efforts enabled the dissection of morphological and electrical parameters that affect differences in EPSPs velocity and latency in human versus rat L2/3 PCs' dendrites. We first assumed uniform cable properties for all cells modeled (*Figure 6*), showing that the key parameter affecting the enhanced velocity in human neurons is the large increase in conductance load (sink) imposed by the extended basal tree in human PCs. The larger diameter of the apical dendrite in human also contributes, but to a lesser degree, to this effect. Finally, differences in passive cable properties also slightly favor faster signal propagation in humans. Indeed, when the basal tree of human PCs modeled (now with cable parameters fitted per cell) was replaced by the basal tree of rat PCs (and *vice versa*), the interspecies differences diminished – emphasizing again the key impact on signal propagation velocity of the large conductance load at the soma of human L2/3 PCs resulting from the larger basal tree in human PCs (*Figures 7 and 8*).

## Discussion

Emergence of data concerning conserved and divergent features of different mammalian species in the structure and function of the cerebral cortex suggest fundamental similarity across species (*Defelipe, 2011*; *Galakhova et al., 2022*; *Herculano-Houzel, 2011*) with a subset of specialized features documented in the human cortex. A number of these specialized properties, like the increase in the size of individual neurons detected early by *Cajal, 1899*, have far reaching functional consequences and here we identified some compensatory mechanisms which, in turn, are based on additional specialized features. In particular, we studied propagation velocity of both forward (axonal) and backward (dendritic) action potential, as well as of EPSPs in human and rat dendrites. Our experimentally-based models showed that the average membrane time constant of the two species is similar (~11 ms). Yet, EPSPs arising in the apical dendrite at similar distances from the soma have significantly shorter latency in humans. This results primarily from the larger diameter of the apical trunk in humans, but also from the difference in cable properties between the two species.

Detailed compartmental models of 3D reconstructed and biophysically measured L2/3 PCs of human and rat L2/3 PCs enabled us to systematically explore factors affecting EPSPs propagation velocity and latency in apical dendrites of these two species. Since the diameter of the apical dendrite is larger in human, and assuming that all specific cable parameters were identical, a synapse located in the apical tree at a given physical distance from the soma is electrotonically closer (in units $\lambda$) to the soma in human cells. Consequently, the latency of the dendritic-to-soma EPSP latency is expected to be shorter in human apical dendrites. This shorter cable distance of the human synapse (at a given physical distance) has an additional consequence. The velocity of the EPSP peak in dendritic cables is not constant; it is faster near the synapse, converging to a constant value of $2\lambda/\tau$ away from the synapse (see *Agmon-Snir and Segev, 1993*). Therefore, EPSPs that originated at electrotonically closer synapses to the soma fall on the steeper (faster) phase of their velocity curve, implying a shorter latency to the soma. But we found that the key factor affecting the propagation velocity of EPSPs toward the soma is the degree of conductance load (the boundary condition) at the soma. We show that the significantly larger basal tree in human L2/3 cells implies a larger conductance load there and as shown in *Figures 6 and 8*, this enhances EPSP propagation velocity and reduces synaptic latency to the soma (see also *Agmon-Snir and Segev, 1993*). It is important to note that this increased conductance load (increased sink) in human L2/3 neurons (and probably also in other cortical neurons and other neuron types which are larger in human compared to rat) will enhance EPSPs originated also in the basal and not specifically in the apical tree. The intuitive reason for this enhancement is that the large conductance load (the 'leaky end' boundary conditions) more effectively 'steals' the synaptic (axial) current (like water pouring faster into a large pool). The more mathematical intuition is that the large soma (sink) adds fast time constants to the system (see also the related explanation in Figure 4 in *Eyal et al., 2014*).

Additional factors that favor accelerated signal propagation in human L2/3 dendrites are differences in respective specific cable parameters between human and rat (*Figure 7*). Additional factors

that were not fully explored in the present study, such impedance mismatch due to local morphological irregularities at branch points (*Figure 8—figure supplement 2*) and due to dendritic spines might also play a role in affecting signal propagation speed (*Manor et al., 1991*; *Figure 7—figure supplement 1*).

Noteworthy here is that we found that the membrane time constant, $\tau$, is similar in L2/3 PCs of rodents and human implying the preservation of coincidence detection capabilities of dendrites in both species. This is important because coincidence detection in dendrites is a fundamental mechanism for a variety of plasticity mechanisms and computational functions such as directional selectivity, sound localization, and expansion of the dynamic range of sensory processing (*Agmon-Snir et al., 1998*; *Roome and Kuhn, 2018*; *Wang et al., 2000*) and see review in *Hay et al., 2016*.

Multifaceted upscaling of properties found in the human microcircuit is usually considered instrumental in functional enrichment. For example, increase in the number of human supragranular pyramidal cell types compared to the mouse (*Berg et al., 2021*; *Deitcher et al., 2017*; *Mohan et al., 2015*) might help in separating multiple tasks of parallel processing in cortical circuits in and the increased range of synaptic strength in pyramidal output contributes to increased saliency of individual excitatory cells followed by efficient network pattern generation in human (*Szegedi et al., 2016*; *Szegedi et al., 2016*; *Verhoog et al., 2013*). However, increase in the size and in morphological complexity of individual neurons might not follow a simple bigger is better logic, but instead it is rather a double-edged sword when considering cellular and microcircuit level function (*Dalügge and Remy, 2018*; *Fişek and Häusser, 2020*; *London and Häusser, 2005*; *Mohan et al., 2015*; *Spruston et al., 2016*; *Vetter et al., 2001*). On one hand, additional dendritic length can receive a higher number and a more diverse set of inputs contributing to circuit complexity (*Loomba et al., 2022*) and sophistication of dendritic architecture has been reviewed as the site for elaborate subcellular processing (*Beaulieu-Laroche et al., 2021*; *Deitcher et al., 2017*; *Galakhova et al., 2022*; *Gidon et al., 2020*; *Mohan et al., 2015*). On the other hand, signals need to propagate along a longer route through dendritic or axonal trees of increased size. Without compensatory mechanisms, textbook knowledge dictates that longer propagation times and altered waveforms of signals associate with elongated neural processes (*Agmon-Snir and Segev, 1993*; *Buzsáki et al., 2013*; *Jack et al., 1975*; *Laughlin and Sejnowski, 2003*). Our observation that soma-to-soma pyramidal cell synaptic latencies are similar in human and rodent strongly suggest that compensatory mechanisms evolved together with alterations in dendritic structure such as increased thickness of dendritic segments in the human compared to segments equidistant from the soma in the rat. This finding is backed up by earlier experiments showing similar soma-to-soma latencies between presynaptic pyramidal cells and postsynaptic fast spiking interneurons in human and rat (*Szegedi et al., 2016*) and between human and mouse pre-and postsynaptic cells overall in the neocortex (*Campagnola et al., 2022*). Thus, it appears that signals connecting pyramid-to-pyramid and pyramid-to-interneuron cell pairs have an evolutionarily conserved latency and compensation provided by dendritic structure seems precise. Importantly, based on the datasets available, there is no indication of significant over/under-compensation and acceleration/deceleration of soma-to-soma propagation times.

Precision in monosynaptic latencies across species is instrumental in keeping the timeframe relatively stable for circuit plasticity. Research in animal models laid experimental and theoretical foundations and uncovered bewildering multiplicity of mechanisms explaining the induction and maintenance of plasticity in cortical microcircuits (*Bliss and Collingridge, 2019*; *Dan and Poo, 2004*; *Debanne et al., 2019*; *Hebb, 1949*; *Kullmann et al., 2012*; *Malenka and Bear, 2004*; *Markram et al., 2012*). In contrast, plasticity is understudied in human samples both at the cellular and microcircuit level (*Chittajallu et al., 2020*; *Mansvelder et al., 2019*). Spike time dependent plasticity (STDP) is based on the relative timing of pre-and postsynaptic activity (*Caporale and Dan, 2008*; *Feldman, 2012*; *Markram et al., 1997*) and the paramount feature of STDP experiments to date is that minute jitter between pre-and postsynaptic activity results in major changes in synapse strength (*Bi and Poo, 1998*; *Verhoog et al., 2013*). Pioneering STDP studies in human neurons showed a wide temporal STDP window with a reversed STDP curve compared to classic results detected in rodent brain (*Bi and Poo, 1998*; *Verhoog et al., 2013*). Interestingly, dendritic L-type voltage-gated calcium channels were found important in human STDP rules (*Verhoog et al., 2013*), yet our results indicate that dendritic bAP speed is equally influenced by calcium channels in human and rat. However, the faster bAP propagation found here in human PCs is compatible with the shifted STDP rule switch (*Verhoog et al., 2013*)

by allowing postsynaptic somatic action potentials to be generated later yet arriving to dendrites at the same time relative to presynaptic spikes. It remains to be established how altered cable properties reported here interact through a dynamic interplay between potentially human-specific dendritic ion channel distribution and local dendritic regenerative processes in order to achieve the reversed STDP curve in human (*Beaulieu-Laroche et al., 2018*; *Beaulieu-Laroche et al., 2021*; *Dalügge and Remy, 2018*; *Fişek and Häusser, 2020*; *Gidon et al., 2020*; *Kalmbach et al., 2018*).

In addition to associative plasticity, precision of synaptic delays is crucial in the generation of circuit oscillations and network synchronization. Although all known patterns of local field potentials and behavioral correlates present in the human cortex can be detected in other mammals (*Buzsáki et al., 2013*), fast oscillations are thought to be especially important in cognitive performance (*Buzsáki, 2015*; *Klinzing et al., 2019*; *Ward, 2003*). Fast population rhythms in the cerebral cortex in the gamma and high gamma range are based on alternating activation of monosynaptically coupled and reciprocally connected pyramidal cells and interneurons (*Averkin et al., 2016*; *Buzsáki and Wang, 2012*) and similar mechanisms were proposed for some forms of ripple oscillations (*Averkin et al., 2016*; *Komlósi et al., 2012*; *Molnár et al., 2008*). The relatively small axonal distances and accordingly short axonal AP propagation latencies linking locally connected human PCs and or interneurons found here and earlier (*Campagnola et al., 2022*; *Goriounova et al., 2018*; *Komlósi et al., 2012*; *Molnár et al., 2008*; *Molnár et al., 2016*; *Verhoog et al., 2013*) are compatible with the frequency range of fast oscillations. Brief loop times during sequential reactivation of a subset of closely located neurons participating in fast human rhythms are helped by subcellular placement of PC-to-PC (and PC-to-fast spiking interneuron *Molnár et al., 2008*; *Molnár et al., 2016*) synapses on midrange dendritic segments instead of distal branches and by extremely effective glutamatergic synapses on interneurons triggering postsynaptic spikes in response to unitary input from a PC (*Molnár et al., 2008*; *Molnár et al., 2016*) in addition to accelerated human dendritic signal propagation. Indeed, latencies of monosynaptic spike-to-spike coupling in single-cell triggered Hebbian assemblies characteristic to the human cortical circuit are compatible with up to ~200 Hz frequency (*Komlósi et al., 2012*; *Molnár et al., 2008*). Phasic and sequential firing of interneurons and PCs was reported in vivo during fast oscillations in humans (*Le Van Quyen et al., 2016*) and single-cell spiking sequences emerging during human memory formation are replayed during successful memory retrieval (*Vaz et al., 2020*) similar to results pioneered in the hippocampus of rodents (*Nádasdy et al., 1999*; *Skaggs and McNaughton, 1996*; *Wilson and McNaughton, 1994*). Our results suggest that changes in human dendritic properties contribute to cross-species preservation of fast oscillation-related cortical function at the local microcircuit level.

# Materials and methods
## Experimental design
### Slice preparation
Experiments were conducted according to the guidelines of the University of Szeged Animal Care and Use Committee (ref. no. XX/897/2018) and of the University of Szeged Ethical Committee and Regional Human Investigation Review Board (ref. 75/2014). For all human tissue material written consent was given by the patients prior to surgery. Human neocortical slices were sectioned from material that had to be removed to gain access for the surgical treatment of deep-brain target (n = 33 female and n = 29 male, aged 49± 19.2 y, from the frontal (n=21), temporal (n=20), parietal (n=20) and occipital (n=1) cortices). Anesthesia was induced with intravenous midazolam and fentanyl (0.03 mg/kg, 1–2 µg/kg, respectively). A bolus dose of propofol (1–2 mg/kg) was administered intravenously. The patients received 0.5 mg/kg rocuronium to facilitate endotracheal intubation. The trachea was intubated, and the patient was ventilated with $O_2$/$N_2O$ mixture at a ratio of 1:2. Anesthesia was maintained with sevoflurane at care volume of 1.2–1.5. Following surgical removal, the resected tissue blocks were immediately immersed into a glass container filled with ice-cold solution in the operating theater and transported to the electrophysiology lab. For animal experiments, we used the somatosensory cortex of young adults (19–46 d of age, (P) 23.9±4.9) male Wistar rats. Before decapitation animals were anesthetized by inhalation of halothane. 320 µm thick coronal slices were prepared with a vibration blade microtome (Microm HM 650 V; Microm International GmbH, Walldorf, Germany). Slices were cut in ice-cold (4 °C) cutting solution (in mM) 75 sucrose, 84 NaCl, 2.5 KCl, 1 NaH₂PO₄, 25

NaHCO$_3$, 0.5 CaCl$_2$, 4 MgSO$_4$, 25 D(+)-glucose, saturated with 95% O$_2$ and 5% CO$_2$. The slices were incubated in 36 °C for 30 min, subsequently the solution was changed to (in mM) 130 NaCl, 3.5 KCl, 1 NaH$_2$PO$_4$, 24 NaHCO$_3$, 1 CaCl$_2$, 3 MgSO$_4$, 10 D(+)-glucose, saturated with 95% O$_2$ and 5% CO$_2$, and the slices were kept in it until experimental use. The solution used for recordings had the same composition except that the concentrations of CaCl$_2$ and MgSO$_4$ were 3 and 1.5 mM unless it is indicated otherwise. The micropipettes (3–5 MΩ) were filled (in mM) 126 K-gluconate, 4 KCl, 4 ATP-Mg, 0.3 GTP-Na$_2$, 10 HEPES, 10 phosphocreatine, and 8 biocytin (pH 7.25; 300 mOsm).

## In vitro electrophysiology

Somatic whole-cell recordings were obtained at ~37 °C from simultaneously recorded PC-PC cell pairs visualized by infrared differential interference contrast (DIC) video microscopy at depths 60–160 µm from the surface of the slice (Zeiss Axio Examiner LSM7; Carl Zeiss AG, Oberkochen, Germany), 40x water-immersion objective (1.0 NA; Carl Zeiss AG, Oberkochen, Germany) equipped with Luigs and Neumann Junior micromanipulator system (Luigs and Neumann, Ratingen, Germany) and HEKA EPC 10 patch clamp amplifier (HEKA Elektronik GmbH, Lambrecht, Germany). Signals were digitalized at 15 kHz and analyzed with custom-written scripts in Python. Presynaptic cells were stimulated with a brief suprathreshold current paired-pulse (800 pA, 2–3 ms, 50–60 ms separation of the two pulses), derived in 10 s interval. The postsynaptic cells were recorded in current-clamp recording, holding current was set to keep the cell's membrane potential around −50 mV. The experiments were stopped if the series resistance (Rs) exceeded 25 MΩ or changed more than 20%. For the dendritic recordings 20 µM Alexa 594 was added to the internal solution of the somatic pipette and 20 µM Alexa 488 to the internal solution of the dendritic pipette. The PCs were kept in whole-cell configuration for at least 10 min before the axon bleb or dendritic targeted recording started. Then the microscope switched to 2 p mode. The fluorescent dyes of the pipette solution were excited at 850 nm wavelength with a femtosecond pulsing Ti:sapphire laser (Mai Tai DeepSee, Spectra-Physics, Santa Clara, CA). The axon blebs and the dendrites were targeted in 2 p mode. After the successful seal formation, the imaging was switched off to reduce the phototoxicity in the sample. All the recordings were carried out in current clamp mode. 800ms long square pulses with elevating amplitude (from −110–300 pA) were used to evoke APs. In some experiments, the same long square injection protocol was repeated at the dendritic/axonal recording site. For measuring the forward propagation of electrical signals in dendrites, we applied either short artificial EPSC-shaped currents (*Connelly et al., 2016*) or mostly ramp currents (*Markram and Sakmann, 1994*) through the dendritic pipette. Ten minutes of recording we applied different drugs or finished the recordings. At the end of the recording, we acquired a 2 p Z series from the recorded dendrite. Then the pipettes were carefully withdrawn from the cells. The slices went under chemical fixation for further anatomical investigation. Due to the small diameter of the dendrites of L2/3 neurons, the dendritic pipette access resistance was 92.43±34.29 MΩ with 24.8–196.2 MΩ range (*Gidon et al., 2020*). We ran a set of computer simulations on our reconstructed neurons (both human and rat), adding a simulated electrode with variable serial resistance values. We found that, for series resistances ranging from 40 to 200 MΩ, the effect of the presence of the electrode on the EPSP latencies is negligible (*Figure 8—figure supplement 3*).

The specific membrane capacitance recordings were carried out as described previously (*Gentet et al., 2000*). Briefly, the L2/3 PCs were whole cell patch clamped, and a gentle suction made during slow withdrawal of the pipette. The nucleus of the cells were pulled out and the voltage clamped at −70 mV. −5 mV voltage steps (repeated 100 times) were applied and the capacitive transients were measured. A DIC image of the nucleus were made for calculation of the membrane surface with the following equation:

$$A = \frac{(a + b)^2 * \pi}{A} \tag{1}$$

Where a is the shorter diameter of the ellipse and b is the longer one. After the recording the nucleus was blown away and the pipette tip was pushed into a sylgard ball until the GΩ seal formed. The −5 mV voltage steps were applied again to record the residual capacitance of the system. Before the analysis, we subtracted the residual capacitance from the transients.

Pharmacological experiments were carried out on PCs during simultaneous somatic and dendritic recordings after 10 min of control recording using ACSF with the following drugs: 20 µM

4-(*N*-ethyl-*N*-phenylamino)–1,2 dimethyl-6-(methylamino)pyrimidinium chloride (ZD7288) (Sigma-Aldrich), or 1 μM TTX, 200 μM CdCl$_2$, and 20 μM AP5.

## Post hoc anatomical analysis of recorded cell pairs

After electrophysiological recordings, slices were fixed in a fixative containing 4% paraformaldehyde, 15% picric acid, and 1.25% glutaraldehyde in 0.1 M phosphate buffer (PB; pH = 7.4) at 4 °C for at least 12 hr. After several washes in 0.1 M PB, slices were cryoprotected in 10% then 20% sucrose solution in 0.1 M PB. Slices were frozen in liquid nitrogen then thawed in PB, embedded in 10% gelatin, and further sectioned into slices of 60 μm in thickness. Sections were incubated in a solution of conjugated avidin-biotin horseradish peroxidase (ABC; 1:100; Vector Labs) in Tris-buffered saline (TBS, pH = 7.4) at 4 °C overnight. The enzyme reaction was revealed by 3'3-diaminobenzidine tetrahydrochloride (0.05%) as chromogen and 0.01% H$_2$O$_2$ as an oxidant. Sections were post-fixed with 1% OsO$_4$ in 0.1 M PB. After several washes in distilled water, sections were stained in 1% uranyl acetate, dehydrated in an ascending series of ethanol. Sections were infiltrated with epoxy resin (Durcupan, Sigma-Aldrich) overnight and embedded on glass slices. 3D light microscopic reconstructions were carried out using the Neurolucida system with a 100× objective. The number of putative synaptic contacts were determined by searching for close appositions of presynaptic axon terminals and postsynaptic dendrites under light microscopy. The distance of the contact sites alongside the branches were measured with Neurolucida. The intersomatic distance was calculated from the branch length from the presynaptic soma to the putative synaptic contact alongside the axon, and the length of the dendrite from the contact site to the postsynaptic soma. If there were more than one putative synapse between the cells, we took the shortest intersomatic path distance for that given cell pair.

## Electron microscopy

Sample preparations for the electron microscopy were performed as described previously (*Boldog et al., 2018*; *Molnár et al., 2016*). Briefly, digital images of serial EM sections (20 nm thickness) were taken at 64,000 x magnification with a FEI/Philips CM10 electron microscope equipped with a MegaView G2 camera. The membrane thickness measurements were carried out on digital images with a custom software. Briefly, postsynaptic dendritic structures were identified with the presence of postsynaptic densities (PSD) faced in front of axon terminals filled with vesicles. At least 20 nm away from the PSD, perpendicular lines were used as region interests (ROI). The intensity line profile of each ROI was calculated, and edge detection was carried out on them. The thickness was determined as the distance between the first and last point along the line profile where the gradient magnitude was larger than 50.

## Data analysis

The electrophysiological recordings were analyzed by custom-written python scripts. First, the recorded sweeps were exported with HEKA FitMaster to ascii files. The mean synaptic delay in the paired recordings was determined by the averages of the delays between the peak of single presynaptic action potentials and the onsets of the corresponding EPSPs. The onset was determined by the projection of the intersection of two linear fits on the postsynaptic signal (*Fedchyshyn and Wang, 2007*). The first line was fitted to the baseline 1ms window from –0.5 to +0.5 ms of the presynaptic AP peak. The second line was fitted on the rising phase of the EPSP (5–30% of the amplitude). The time point of the crossing lines was projected back to the signal and it was used as the onset (*Figure 1B*). For the forward propagation dendritic experiments the latency was calculated on an average signal. The onset of the EPSP-like waveform was determined as the onset of EPSPs in the paired recordings.

The bAP latency was measured at the peak of the average signal for each cell (*Stuart and Sakmann, 1994*). Only the first APs of the sweeps were averaged to avoid activity dependent Na$^+$ channel inactivation that can cause a putative modulatory effect on the signal propagation speed. For the axon bleb recordings we assumed that the axon initial segment (AIS) of the cells are 35 μm from the axon hillock (*Palmer and Stuart, 2006*), and the APs propagate forward (to the bleb) and backward (to the soma) at the same speed. For the correction of the AIS we used the following formula:

$$vcorr = \frac{l}{t + (ais/l * t)} \qquad (2)$$

where *vcorr* is the corrected propagation speed for AIS position, *l* is the axonal distance between the soma and the axon bleb, *t* is the latency between the two measuring point, *ais* is the assumed position of the AIS alongside the axon (35 μm).

## Estimating passive parameters of L2/3 pyramidal cells

We constructed detailed passive compartmental and cable models for five L2/3 human neurons and the four rat L2/3 neurons that were both 3D morphologically reconstructed and biophysically characterized. For each modeled neuron, we optimized the values of three key passive parameters: the specific membrane resistivity and capacitance ($R_m$, $C_m$) and the specific axial resistivity, $R_a$, using Neuron 8.0 (*Hines et al., 2009*) principal axis optimization algorithm (*Brent, 1976*; *Segev et al., 1989*). Optimization was achieved by minimizing the difference between experimental voltage response following hyperpolarizing current steps either to the soma or to the apical dendrite (*Figure 7A and B*) and the model response. When possible, experimental data was averaged over repetitions of the same stimulus.

To account for the surface area of spines, we used the spine correction factor (F) of 1.9 and 1.5 for human and rat PCs, respectively, by multiplying $C_m$ and dividing $R_m$ by F in segments which are at a distance of at least 60 μm from the soma (*Eyal et al., 2016*; *Hunt et al., 2023*). In this study, we did not attempt to fit the nonlinear effect of $I_h$ of the voltage response of the cells.

As our experimental data contains simultaneous soma-dendritic pair recordings/stimulation, we decided to fit the voltage response in one location (e.g. the soma) for the current injection in the other location (e.g. dendrites). This is a cleaner way compared to the typical case when only one recording/ stimulating electrode is available, as the problem of bridge balance at the current input site does not exist in this case. As we have two recording and simulation sites, we also examined how well the model predicts the local voltage response at the injection site (*Figure 7B*). Analysis and simulation were conducted using Python 3.8 and visualization using matplotlib 3.15 (*Hunter, 2007*) and seaborn 0.11 (*Waskom, 2021*).

## Modeling EPSP propagation delay and velocity

We used the NEURON simulator (*Hines et al., 2009*) to model the nine 3D reconstructed neurons (*Figure 6—figure supplement 1*). To compute EPSP's propagation latency and velocity, we simulated EPSPs by injecting a brief transient alpha-shaped current, $I_\alpha$, delivered either to the soma or in various dendritic loci along the modeled apical tree.

$$I_\alpha = A \left( 1 - e^{\frac{-t}{\tau_0}} \right) - \left( 1 - e^{\frac{-t}{\tau_1}} \right) \tag{3}$$

where $A = 1.5$, $\tau_0 = 0.25$ and $\tau_1 = 1ms$, resulting in EPSP peak time, $t_{peak} = 0.5ms$ and peak current of $I_{peak} = 1.4nA$.

Latency of the resultant EPSP was calculated as the difference between the EPSP peak at all dendritic branches and its resulting EPSP at the soma; using a sampling time bin of 0.01ms. Velocity was calculated as the distance of the input site from soma divided by latency between these two points. Each dot in *Figures 6 and 7* is the respective value for a specific dendritic segment in a specific branch of a neuron's apical tree. For each measured feature (radius, and velocity), an inset (zoom-in) matching the experimental distance range was added. It shows the average value across dendritic branches with a given distance from the soma, as a function of distance from soma, smoothed with a rolling 10 μm window. For normalizing the path distance of a given dendritic site to the soma in cable units, we calculated the space constant

$$\lambda = \sqrt{d \frac{R_m}{4} R_a} \tag{4}$$

for each dendritic segment (where d is the segment's diameter). We then summed up the cable lengths of all segments along the path from the dendritic location to the soma. Time was normalized by the membrane time constant $\tau = R_m * C_m$. Note that, for segments far enough from cable boundary conditions and stimulus location, velocity approximately equals the theoretical value of $2\lambda / \tau$, (*Agmon-Snir and Segev, 1993*) see *Figure 7—figure supplement 1*. Hence, in the uniform case where all

specific parameters are equal for all cell modeled (*Figure 6*), normalizing the distance in cable should equalize latency/velocity differences resulting from diameter differences.

To account for brain tissue shrinkage due to fixation, for every segment, diameter and length were scaled based on an estimation of specific neuron shrinkage (see *Supplementary file 1*). To account for unequal dye spread, for a few manually picked segments, diameter value was fixed to be equal to its nearby segment (to avoid sudden diameter jump).

## Equivalent cables for human and rat L2/3 PCs

'Equivalent cables' for the respective 9 modeled human and rat cells was based on Rall's cable theory (*Rall, 1959*). The variable diameter, $d_{eq}(X)$, of this cable, as seen from the soma is,

$$d_{eq}(X) = \left( \sum_j d_j(X)^{\frac{3}{2}} \right)^{\frac{2}{3}} \tag{5}$$

where X is the cable (electrotonic) distance from the soma and $d_j(X)$ is the diameter of the j$^{th}$ dendrite at the distance X from that point of interest. *Figure 8A* shows such equivalent cables as seen from the soma. The equivalent cable for the basal tree is depicted in red and for the apical tree in blue. This enables one to graphically appreciate the large difference in the conductance load (current sink) imposed by basal tree between human and rat L2/3.

## Statistical analysis

Statistical analyses were performed in Python v.3.6, using the Python packages DABEST (*Ho et al., 2019*), scipy, numpy, matplotlib (*Hunter, 2007*), seaborn (*Waskom, 2021*), pandas, pinguin (*Vallat, 2018*), bioinfokit and scikit-learn. SHAP (*Lundberg and Lee, 2017*) and GLM (*Kiebel and Holmes, 2007*) models were done with shap python package with scikit-learn Gradient Boosting Regressor (*Friedman, 2002*) and with statsmodels.glm with gamma family. Interaction formula: latency ~species + distance + radius + (species × sfitted × distance) + (species × hybrid × distance) + (species ×radius). No interaction: latency ~ species +distance + hybrid +radius + fitted.

Data presented as the mean ± s.d. Normality was tested with the Shapiro-Wilk test. For statistical analysis, t-test, Mann-Whitney U-test or Wilcoxon signed-rank test was used. For the statistical analysis of the drug treatments, we used two ways ANOVA with repeated measures and Tukey HSD test for posthoc comparisons. Correlations were tested using Pearson's correlation, respectively. We used the Gardner-Altman estimation plot throughout this study which is a bootstrap-coupled estimation of effect sizes, plotting the data against a mean (paired mean, as indicated) difference between the leftmost condition and one or more conditions on the right (right y-axis), and compared this difference against zero using 5000 bootstrapped resamples. In these estimation graphics, each black dot indicates a mean difference, and the associated black ticks depict error bars representing 95% confidence intervals; the shaded area represents the bootstrapped sampling-error distribution (*Ho et al., 2019*). Differences were accepted as significant if p<0.05. The complete results of all the statistical analysis presented on the main figures and figure supplements can be found as supplementary files.

## Acknowledgements

The authors thank Éva Tóth, Katalin Kocsis, Leona Mezei and Bettina Lehóczki for assistance in anatomical experiments, Judith Baka for providing the electron micrographs for membrane thickness measurements, Gergely Komlósi, Martin Tóth, Miklós Füle, Szabina Furdan, Szabolcs Oláh, Zoltán Péterfi for recording some neuron and Attila Ozsvár, Márton Rózsa, Martin Tóth, Ildikó Szöts, Norbert Mihut, Róbert Averkin, Sándor Bordé, Viktor Szegedi for useful feedback and suggestions. The technical help and methodical suggestions of János Szabadics, János Brunner, and Viktor Oláh at the beginning of the project are appreciated. This work is dedicated to the memory of Mrs. Lily Safra, a great supporter of brain research. Funding: This work was supported by Hungarian Research Network grants HUN-REN-SZTE Agykérgi Neuronhálózatok Kutatócsoport and KÖ–36/2021 (G T). Ministry of Human Capacities Hungary (20391-3/2018/FEKUSTRAT and NKP 16–3-VIII-3) (G T); National Research, Development and Innovation Office grants GINOP 2.3.2-15-2016-00018, Élvonal KKP 133807, ÚNKP-20–5 - SZTE-681, 2019–2.1.7-ERA-NET-2022–00038, TKP2021-EGA-09, TKP-2021-EGA-28 (G T) and OTKA

K128863 (G T, G M). ÚNKP-21–5-SZTE-580 New National Excellence Program of the Ministry for Innovation and Technology from the source of the National Research, Development and Innovation Fund (G M). ÚNKP 16–3-VIII-3 new national excellence program of the Ministry of Human Capacities (G O). János Bolyai Research Scholarship of the Hungarian Academy of Sciences (G M). Hungarian Scientific Research Foundation under grant ANN-135291 (A Sz). National Institutes of Health awards U01MH114812 (G T., I S) and UM1MH130981 (G T). The Patrick and Lina Drahi Foundation, grant from the ETH domain for the Blue Brain Project, the Gatsby Charitable Foundation (I S).

## Additional information

### Funding

| Funder | Grant reference number | Author |
| --- | --- | --- |
| National Research, Development and Innovation Office | GINOP 2.3.2-15-2016-00018 | Gábor Tamás |
| National Research, Development and Innovation Office | ÚNKP-20-5 - SZTE-681 | Gábor Tamás |
| National Research, Development and Innovation Office | TKP2021-EGA-09 | Gábor Tamás |
| National Research, Development and Innovation Office | TKP-2021-EGA-28 | Gábor Tamás |
| National Research, Development and Innovation Office | OTKA K128863 | Gábor Molnár<br>Gábor Tamás |
| Ministry for Innovation and Technology | ÚNKP-21-5-SZTE-580 | Gábor Molnár |
| Ministry of Human Capacities | ÚNKP 16-3-VIII-3 | Gábor Tamás |
| Hungarian Academy of Sciences | János Bolyai Research Scholarship | Gábor Molnár |
| Hungarian Academy of Sciences | ANN-135291 | Attila Szücs |
| National Institutes of Health | U01MH114812 | Idan Segev<br>Gábor Tamás |
| National Institutes of Health | UM1MH130981 | Gábor Tamás |
| The Patrick and Linda Drahi Foundation | | Idan Segev |
| ETH domain for the Blue Brain Project | | Idan Segev |
| the Gatsby Charitable Foundation | | Idan Segev |
| Hungarian Research Network (HUN-REN-SZTE Agykérgi Neuronhálózatok Kutatócsoport) | | Gábor Tamás |
| National Research, Development and Innovation Office | Élvonal KKP 133807 | Gábor Tamás |

| Funder | Grant reference number | Author |
|---|---|---|
| National Research, Development and Innovation Office | 2019-2.1.7-ERA-NET-2022-00038 | Gábor Tamás |
| Ministry of Human Capacities Hungary | 20391-3/2018/FEKUSTRAT | Gábor Tamás |
| Ministry of Human Capacities Hungary | NKP 16–3-VIII-3 | Gábor Tamás |
| Hungarian Research Network | KÖ-36/2021 | Gábor Tamás |

The funders had no role in study design, data collection and interpretation, or the decision to submit the work for publication.

### Author contributions

Gáspár Oláh, Conceptualization, Software, Formal analysis, Investigation, Visualization, Methodology, Writing – original draft, Writing – review and editing; Rajmund Lákovics, Yonatan Leibner, Formal analysis, Investigation, Methodology; Sapir Shapira, Software, Formal analysis, Investigation, Visualization, Methodology, Writing – original draft, Writing – review and editing; Attila Szücs, Software, Formal analysis, Investigation, Methodology; Éva Adrienn Csajbók, Pál Barzó, Investigation, Methodology; Gábor Molnár, Conceptualization, Formal analysis, Supervision, Investigation, Visualization, Methodology, Writing – original draft, Writing – review and editing; Idan Segev, Gábor Tamás, Conceptualization, Formal analysis, Supervision, Funding acquisition, Methodology, Writing – original draft, Writing – review and editing

### Author ORCIDs

Rajmund Lákovics ⓘ https://orcid.org/0000-0001-7261-2522
Attila Szücs ⓘ https://orcid.org/0000-0001-9733-4135
Gábor Molnár ⓘ https://orcid.org/0000-0001-7959-139X
Idan Segev ⓘ https://orcid.org/0000-0001-7279-9630
Gábor Tamás ⓘ https://orcid.org/0000-0002-7905-6001

### Ethics

Experiments were conducted according to the guidelines of University of Szeged Animal Care and Use Committee (ref. no. XX/897/2018) and of the University of Szeged Ethical Committee and Regional Human Investigation Review Board (ref. 75/2014). For all human tissue material written consent was given by the patients prior to surgery.

Experiments were conducted according to the guidelines of University of Szeged Animal Care and Use Committee (ref. no. XX/897/2018) and of the University of Szeged Ethical Committee and Regional Human Investigation Review Board (ref. 75/2014).

Reviewer #1 (Public review): https://doi.org/10.7554/eLife.93781.4.sa1
Reviewer #2 (Public review): https://doi.org/10.7554/eLife.93781.4.sa2
Reviewer #3 (Public review): https://doi.org/10.7554/eLife.93781.4.sa3
Author response https://doi.org/10.7554/eLife.93781.4.sa4

## Additional files

### Supplementary files

Supplementary file 1. Morphological scaling factors due to fixation.

Supplementary file 2. Model prediction of the average EPSPs latency within experimental recording distance range per modeled cell for the case of identical cable parameters for all cells. $l_{avg}$ is the average physical distance from which the respective experiments (per cell) were performed (zoom-in region in *Figure 6A and B*). $d_{avg}$ is the average diameter at the same range. $L_{avg}$ is the respective distances in cable units ($L = \frac{l}{\lambda}$). Latency is the average latency measured at the experimental distance. Uniform cable parameters were used for all cells as in *Figure 6*.

Supplementary file 3. Model prediction of the average excitatory postsynaptic potentials (EPSPs) latency within experimental recording distance range per modeled cell for the case of identical cable parameters and 'hybrid cell' whereby all modeled cells consist of the basal tree of 'Rat4'. $l_{avg}$ is the average physical distance from which the respective experiments (per cell) were performed (zoom-in region in **Figure 6E and F**). $d_{avg}$ is the average diameter at the same range. $L_{avg}$ is the respective distances in cable units ($L = \frac{l}{\lambda}$). Latency is the average latency measured at the experimental distance. Uniform cable parameters were used for all cells as in **Figure 6**.

Supplementary file 4. Model prediction of the average EPSPs latency within experimental recording distance range per modeled cell for the case of fitted cable parameters per cell and 'hybrid cell' where all modeled cells consist of 'Rat4' basal tree. $l_{max}$ is the maximal physical distance from which the respective experiments (per cell) were performed (zoom-in region in **Figure 8—figure supplement 2** A and B). $d_{max}$ is the (average) diameter at $l_{max}$. $L_{max}$ is the respective distances in cable units ($L = \frac{l}{\lambda}$); $\tau$ is the membrane time constant ($C_m * R_m$). Latency is the average latency measured at the maximal distance. Fitted cable parameters were used for all cells as in **Figure 7**.

Supplementary file 5. Examining factors influencing excitatory postsynaptic potential (EPSP) latency via GLM model without interaction terms. Note the ranking (1-5) of the factors affecting EPSP latency (in ms units). Factors were ranked (1-5) based on the magnitude of their absolute coefficients, which provide insight into their relative contribution to the model. The model was fit using the Gamma family and included continuous factors that were standardized prior to fitting, as well as categorical factors (see comments above for the reference values). Each factor used in the model's formula is highlighted in bold. The formula used for the model: latency ~species + distance + hybrid + radius + fitted.

Supplementary file 6. Examining factors influencing excitatory postsynaptic potentials (EPSP) latency via GLM model with interaction terms. This table presents evidence of significant interactions between the main factors affecting EPSP latency (see **Supplementary file 5** for the non-interaction model). The model was fit using the Gamma family and included continuous factors that were standardized prior to fitting, as well as categorical factors (see **Supplementary file 5** for the reference value). The formula used for the model (names match the factors from **Supplementary file 5**): latency ~species + distance + radius + (species × fitted × distance) + (species × hybrid × distance) + (species × radius).

MDAR checklist

## Data availability

All recordings are available at https://doi.org/10.5281/zenodo.13913084. Analysis codes for experimental physiology are available at: https://github.com/GasparOlah/Analysis_scripts_for_Accelerated-signal-propagation-speed-in-human-neocortical-dendrites (copy archived at **Oláh, 2025**). Codes for modelling and related analyses are available at https://github.com/ssapir/AcceleratedSignalPaper (copy archived at **Shapira, 2025**).

The following dataset was generated:

| Author(s) | Year | Dataset title | Dataset URL | Database and Identifier |
|---|---|---|---|---|
| Oláh G, Lákovics R, Shapira S, Leibner Y, Szücs A, Barzó P, Molnár G, Segev I, Tamás G | 2024 | Accelerated signal propagation speed in human neocortical dendrites | https://doi.org/10.5281/zenodo.13913084 | Zenodo, 10.5281/zenodo.13913084 |

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
