## [Editor Report · eLife Assessment]

This study provides **valuable** observations indicating that human pyramidal neurons propagate information as fast as rat pyramidal neurons despite their larger size. **Convincing** evidence demonstrates that this property is due to several biophysical properties of human neurons. This study will be of interest to neurophysiologists.

---

## [Referee Report · Reviewer #1 (Public review)]

The propagation of electrical signals within neuronal circuits is tightly regulated by the physical and molecular properties of neurons. Since neurons vary in size across species, the question arises whether propagation speed also varies to compensate for it. The present article compares numerous speed-related properties in human and rat neurons. They found that the larger size of human neurons seems to be compensated by a faster propagation within dendrites but not axons of these neurons. The faster dendritic signal propagation was found to arise from wider dendritic diameters and greater conductance load in human neurons. In addition, the article provides a careful characterization of human dendrites and axons, as the field has only recently begun to characterize post-operative human cells. There are only a few studies reporting dendritic properties and these are not all consistent, hence there is added value of reporting these findings, particularly given that the characterization is condensed in a compartmental model.

Strengths

The study was performed with great care using standard techniques in slice electrophysiology (pharmacological manipulation with somatic patch-clamp) as well as some challenging ones (axonal and dendritic patch-clamp). Modeling was used to parse out the role of different features in regulating dendritic propagation speed. The finding that propagation speed varies across species is novel as previous studies did not find a large change in membrane time constant nor axonal diameters (a significant parameter affecting speed). A number of possible, yet less likely factors were carefully tested (Ih, membrane capacitance). The main features outlined here are well known to regulate speed in neuronal processes. The modeling was also carefully done to verify that the magnitude of the effects is consistent with the difference in biophysical properties. Hence, the findings appear very solid to me.

Weaknesses

The role of diameter in regulating propagation speed is well known in the axon literature.

Comment on the revised version: the authors have now made clearer that the role of diameter was well known in the manuscript.

---

## [Referee Report · Reviewer #2 (Public review)]

Summary:

In this paper, Oláh and colleagues introduce new research data on the cellular and biophysical elements involved in transmission within the pyramidal circuits of the human neocortex. They gathered a comprehensive set of patch-clamp recordings from human and rat pyramidal neurons to compare how the temporal aspect of neuronal processing is maintained in the larger human neocortex. A range of experimental techniques have been used, including two-photon guided dual whole-cell recordings, electron microscopy, complemented by theoretical and computational methods.

The authors find that synaptically connected pyramidal neurons within the human neocortex have longer intercellular path lengths. They go on to show that the short soma to soma latencies is not due to propagation velocity along the axon but instead reflects a higher propagation speed of synaptic potentials from dendrite to soma. Next, in a series of extensive computational modeling studies focusing on the synaptic potentials, the authors show that the shorter latency may be explained by larger diameters, affecting the cable properties and resulting is relatively faster propagation of EPSPs in the human neuron. The manuscript is well-written, and the physiological experiments and in-depth theoretical steps for the simulations are clear. Whether passive cable properties of the dendrites alone are responsible for higher velocities remains to be further investigated. Based on the present data the contribution of active membrane properties cannot be excluded.

Strengths:

The authors used complex 2P-guided dual whole-cell recordings in human neurons. In combination with detailed reconstructions, these approaches represent the next steps in unravelling the information processing in human circuits.

The computational modelling and cable theory application to the experimentally constrained simulations provides an integrated view of the passive membrane properties of human neurons.

Weaknesses:

Whether the cable properties alone are the main explanation for speeding the electrical signaling in human pyramidal neurons deserves further studies.

---

## [Referee Report · Reviewer #3 (Public review)]

Summary:

This study indicates that connections across human cortical pyramidal cells have identical latencies despite a larger mean dendritic and axonal length between somas in human cortex. A precise demonstration combining detailed electrophysiology and modeling, indicates that this property is due to faster propagation of signals in proximal human dendrites. This faster propagation is itself due to a slightly thicker dendrite, to a larger capacitive load, and to stronger hyperpolarizing currents. Hence, the biophysical properties of human pyramidal cells are adapted such that they do not compromise information transfer speed.

Strengths:

The manuscript is clear and very detailed. The authors have experimentally verified a large number of aspects that could affect propagation speed and have pinpointed the most important one. This paper provides an excellent comparision of biophysical properties between rat and human pyramidal cells. Thanks to this approach a comprehensive description of the mechanisms underlying the acceleration of propagation in human dendrite is provided.

Weaknesses:

The weaknesses I had identified have been addressed by the authors.

---

## [Author Response]

The following is the authors’ response to the previous reviews.

We are grateful for the positive evaluation of the work and the critical points raised by the reviewers. We thank all reviewers for their excellent comments. We believe that these revisions have significantly improved the quality of our study.

In response to the 2nd reviewer, we apologise for the missing data, we failed to provide a P-value of the RM ANOVA post-hoc test, we are very grateful that this was brought to our attention. We have revised the RM ANOVA by using the Tukey HSD post-hoc test, which is generally recommended for pairwise comparisons as it is more robust to unequal sample sizes. The controversial statistical analysis of the overall comparison of speed differences was deleted, as were three supplementary figures (Fig. S4, Fig. S9 and S10), which are less informative in support of the manuscript.